# The AAA protein Msp1 mediates clearance of excess tail-anchored proteins from the peroxisomal membrane

**Nicholas R Weir, Roarke A Kamber, James S Martenson, Vladimir Denic\***

Department of Molecular and Cellular Biology, Harvard University, Cambridge, United States

**Abstract** Msp1 is a conserved AAA ATPase in budding yeast localized to mitochondria where it prevents accumulation of mistargeted tail-anchored (TA) proteins, including the peroxisomal TA protein Pex15. Msp1 also resides on peroxisomes but it remains unknown how native TA proteins on mitochondria and peroxisomes evade Msp1 surveillance. We used live-cell quantitative cell microscopy tools and drug-inducible gene expression to dissect Msp1 function. We found that a small fraction of peroxisomal Pex15, exaggerated by overexpression, is turned over by Msp1. Kinetic measurements guided by theoretical modeling revealed that Pex15 molecules at mitochondria display age-independent Msp1 sensitivity. By contrast, Pex15 molecules at peroxisomes are rapidly converted from an initial Msp1-sensitive to an Msp1-resistant state. Lastly, we show that Pex15 interacts with the peroxisomal membrane protein Pex3, which shields Pex15 from Msp1-dependent turnover. In sum, our work argues that Msp1 selects its substrates on the basis of their solitary membrane existence.

DOI: https://doi.org/10.7554/eLife.28507.001

## Introduction

Tail-anchored (TA) proteins are integral membrane proteins with a single C-terminal transmembrane segment (TMS). In the budding yeast *Saccharomyces cerevisiae,* the majority of TA proteins are captured post-translationally by cytosolic factors of the conserved Guided Entry of TA proteins (GET) pathway, which deliver them to the endoplasmic reticulum (ER) membrane for insertion by a dedicated insertase (*Denic et al., 2013*; *Hegde and Keenan, 2011*). TA proteins native to the outer mitochondrial and peroxisomal membranes are directly inserted into these membranes by mechanisms that are not well defined (*Chen et al., 2014a*; *Papić et al., 2013*, and reviewed in *Borgese and Fasana, 2011*). Gene deletions of GET pathway components (*getΔ*) result in reduced cell growth, TA protein mistargeting to mitochondria, and cytosolic TA protein aggregates (*Jonikas et al., 2009*; *Schuldiner et al., 2008*). Two recent studies identified the ATPase associated with diverse cellular activities (AAA ATPase) Msp1 as an additional factor for supporting cell viability in the absence of GET pathway function (*Chen et al., 2014b*; *Okreglak and Walter, 2014*). Specifically, they observed that *msp1Δ* cells accumulate mislocalized TA proteins in the mitochondria and that double *msp1Δ getΔ* cells have synthetic sick genetic interactions. This sick phenotype is associated with disruption of mitochondrial function and is exacerbated by overexpression of TA proteins prone to mislocalization (*Chen et al., 2014b*). Msp1 is a cytosolically-facing transmembrane AAA ATPase which resides on both mitochondria and peroxisomes (*Chen et al., 2014b*; *Okreglak and Walter, 2014*). Closely-related members of Msp1's AAA ATPase subfamily form hexamers that bind hydrophobic membrane substrates and use the energy of ATP hydrolysis to extract them from the membrane for protein degradation (*Olivares et al., 2016*). Several lines of evidence are consistent with the working model that Msp1 operates by a similar mechanism: ATPase-dead mutations of

**\*For correspondence:** vdenic@ mcb.harvard.edu

**Competing interests:** The authors declare that no competing interests exist.

**eLife digest** The phrase "finding a needle in a haystack" refers to the difficulty of locating a specific target among a large number of very similar objects. Living cells face a comparable challenge whenever they carry out seek and destroy missions aimed at broken or otherwise undesirable molecules. Scientists are still figuring out how these quality control systems can quickly and accurately pick out the few unwanted molecules that occasionally appear in crowds of normal molecules.

Msp1 is a quality control protein that resides on the outer surfaces of two compartments within cells: mitochondria and peroxisomes. Previous work showed that when a protein called Pex15, which is normally found in peroxisomes, is mistakenly sent to mitochondria it is rapidly eliminated by Msp1.

Weir et al. set out to understand if Msp1 can distinguish incorrectly localized Pex15 from correctly localized Pex15. Fluorescence microscopy was used to watch how Msp1 eliminates Pex15 from compartments within living yeast cells. Although Msp1 did not normally recognize Pex15 at peroxisomes, when Weir et al. attempted to over-load peroxisomes with Pex15 they saw that Msp1 provided a counterforce. Comparing how quickly cells eliminated excess Pex15 at peroxisomes with predictions from mathematical models showed that Pex15 "matures" from an Msp1-sensitive to an Msp1-insensitive state. Further experiments revealed that Pex15 binds to another protein found in peroxisomes, called Pex3, which protects Pex15 from Msp1. By contrast, occasional Pex15 molecules that reached mitochondria remained immature and sensitive to Msp1.

Proteins similar to Msp1 are also found in humans, and Weir et al. hope that a better understanding of how Msp1 works in yeast will help scientists studying human disorders caused by defects in similar quality control systems. This could help to combat disease like cancer, neurodegenerative diseases and cystic fibrosis – which have all been linked to quality control systems that have started to target too few or too many proteins.

DOI: https://doi.org/10.7554/eLife.28507.002

Msp1 are unable to complement *msp1Δ* mutant phenotypes; mitochondrial mistargeting of TA proteins leads to their enhanced co-immunoprecipitation with ATPase-dead Msp1; cells lacking Msp1 have increased half-lives of mistargeted TA proteins; and lastly, a complementary analysis of the mammalian Msp1 homolog ATAD1 (*Chen et al., 2014b*) established a conserved role for Msp1 in correcting errors in TA protein sorting.

Substrate selectivity mechanisms of many AAA proteins have been successfully dissected by bulk cell approaches for measuring substrate turnover. These approaches are resolution-limited, however, when used to study Msp1 in *getΔ* cells because TA proteins mistargeted to mitochondria co-exist with a dominant TA population that remains correctly localized in the same cell. Previous studies overcame this issue through two different approaches that increased the ratio of mistargeted to properly localized substrates. In one case, cells were engineered to produce a Pex15 deletion mutant (Pex15$_{\Delta C30}$) that is efficiently mistargeted to mitochondria because it lacks its native peroxisomal targeting signal (*Okreglak and Walter, 2014*). A major limitation of this approach, however, is its inherent unsuitability for establishing if native Pex15 is a latent Msp1 substrate because of undefined peroxisomal factors. Second, a cell microscopy pulse-chase approach was used to monitor turnover of mitochondrial signal from transiently expressed fluorescently-labeled wild-type Pex15 made susceptible to mistargeting by deletion of *GET3* (*Chen et al., 2014b*). In this approach, expression of Pex15 was transcriptionally controlled by the inducible *GAL* promoter in cells expressing wild-type, ATPase-dead, or no Msp1. Comparison of mitochondrial Pex15 clearance following *GAL* promoter shut-off revealed that cells lacking functional Msp1 had a reduced fractional rate of substrate clearance (*Chen et al., 2014b*); however, these cells also had a larger starting population of mitochondrial Pex15. Thus the presence of Msp1 during Pex15 pulse periods (*Chen et al., 2014b*; *Okreglak and Walter, 2014*) leaves open the possibility that Msp1 does not mediate substrate extraction from the mitochondrial outer membrane but instead blocks substrate insertion into this membrane. Distinguishing between these possibilities requires better tools for temporally controlling and accurately measuring Msp1 activity in cells.

Substrate recognition by AAA proteins can be controlled by a variety of intrinsic substrate determinants and extrinsic factors (*Olivares et al., 2016*). Some insight into Msp1 substrate selectivity comes from negative evidence showing that native mitochondrial TA proteins are inefficient Msp1 substrates (*Chen et al., 2014b*). Thus, substrates might contain intrinsic Msp1 recognition determinants or native mitochondrial TA proteins might be protected from Msp1 recognition by extrinsic mitochondrial factors. Similarly, the potential existence of extrinsic peroxisomal factors might explain why Pex15 (a native peroxisomal TA protein) appears to stably co-reside with Msp1 at peroxisomes but is a substrate for Msp1 at mitochondria (*Chen et al., 2014b*; *Okreglak and Walter, 2014*).

## Results

### Efficient clearance of a fully-integrated substrate from mitochondria by de novo Msp1 induction

To generate a defined Msp1 substrate population prior to initiation of Msp1 activity, we utilized two established synthetic drug-inducible gene expression systems to orthogonally control expression of Pex15 and Msp1. Briefly, we created a yeast strain genetic background with two transcriptional activator-promoter pairs: 1. the doxycycline (DOX)-activated reverse tetracycline trans-activator (rTA) (*Roney et al., 2016*) for controlling expression of fluorescently-labeled Pex15 (YFP-Pex15) from the *TET* promoter; and 2. the β-estradiol-activated synthetic transcription factor Z4EV (*McIsaac et al., 2013*) for controlling Msp1 expression from the Z4EV-driven (*ZD*) promoter (*Figure 1—figure supplement 1A–C*). Next, we pre-loaded mitochondria with Pex15 in the absence of any detectable Msp1 (*Figure 1—figure supplement 1A*) by growing cells for 2 hr in the presence of a high DOX concentration (50 μg/ml) necessary to induce sufficient mitochondrial mistargeting (*Figure 1A* and see below). This was followed by 2 hr of DOX wash-out to allow for mitochondrial maturation of newly-synthesized YFP-Pex15 (*Figure 1A*). Using confocal microscopy, we could resolve the relatively faint mitochondrial YFP fluorescence from the much brighter punctate YFP fluorescence (corresponding to peroxisomes, see below) by signal co-localization with Tom70-mTurquoise2 (a mitochondrial marker; *Figure 1B*) (see *Figure 1—figure supplement 2*, *Videos 1* and *2*, and Materials and methods for computational image analysis details). Lastly, we monitored changes in mitochondrial YFP-Pex15 fluorescence density by timelapse live-cell imaging in the presence or absence of β-estradiol to define the effect of de novo induction of Msp1 activity (*Figure 1A*). Starting with the same pre-existing mitochondrial Pex15 population, we found that de novo Msp1 induction significantly enhanced mitochondrial YFP signal decay (*Figure 1B–C*). We reached a similar conclusion when we used a deletion variant of Pex15 (Pex15$_{\Delta C30}$) that is efficiently mistargeted to mitochondria because it lacks a C-terminal peroxisome targeting signal (*Okreglak and Walter, 2014*)(*Figure 2A–C*). To establish if Pex15$_{\Delta C30}$ was fully membrane-integrated prior to Msp1 induction, we harvested cells after DOX treatment. Following cell lysis, we isolated crude mitochondria by centrifugation and treated them with Proteinase K (PK). Immunoblotting analysis against a C-terminal epitope engineered on Pex15 revealed the existence of a protected TMS-containing fragment that became PK-sensitive after solubilizing mitochondrial membranes with detergent (*Figure 2D*). Taken together, these findings argue that Msp1 can extract a fully-integrated substrate from the mitochondrial outer membrane and gave us a new tool for mechanistic dissection of Msp1 function in vivo.

### Differential kinetic signatures of mitochondrial versus peroxisomal Pex15 clearance by Msp1

While performing the previous analysis, we observed that β-estradiol also enhanced YFP-Pex15 signal decay at punctate, non-mitochondrial structures. To test if these punctae corresponded to peroxisomes, we used a strain with mCherry-marked peroxisomes (mCherry-PTS1) and induced YFP-Pex15 expression with a lower DOX concentration (10 μg/ml). Indeed, we saw robust YFP and mCherry signal co-localization with little apparent Pex15 mistargeting to mitochondria (*Figure 3A–B*). As we initially surmised, β-estradiol-driven Msp1 expression enhanced YFP-Pex15 signal decay at peroxisomes (*Figure 3A–C*). Immunoblotting analysis of lysates prepared from comparably-treated cells provided further support for our conclusion that de novo induction of Msp1 activity enables degradation of peroxisomal Pex15 (*Figure 3D*).

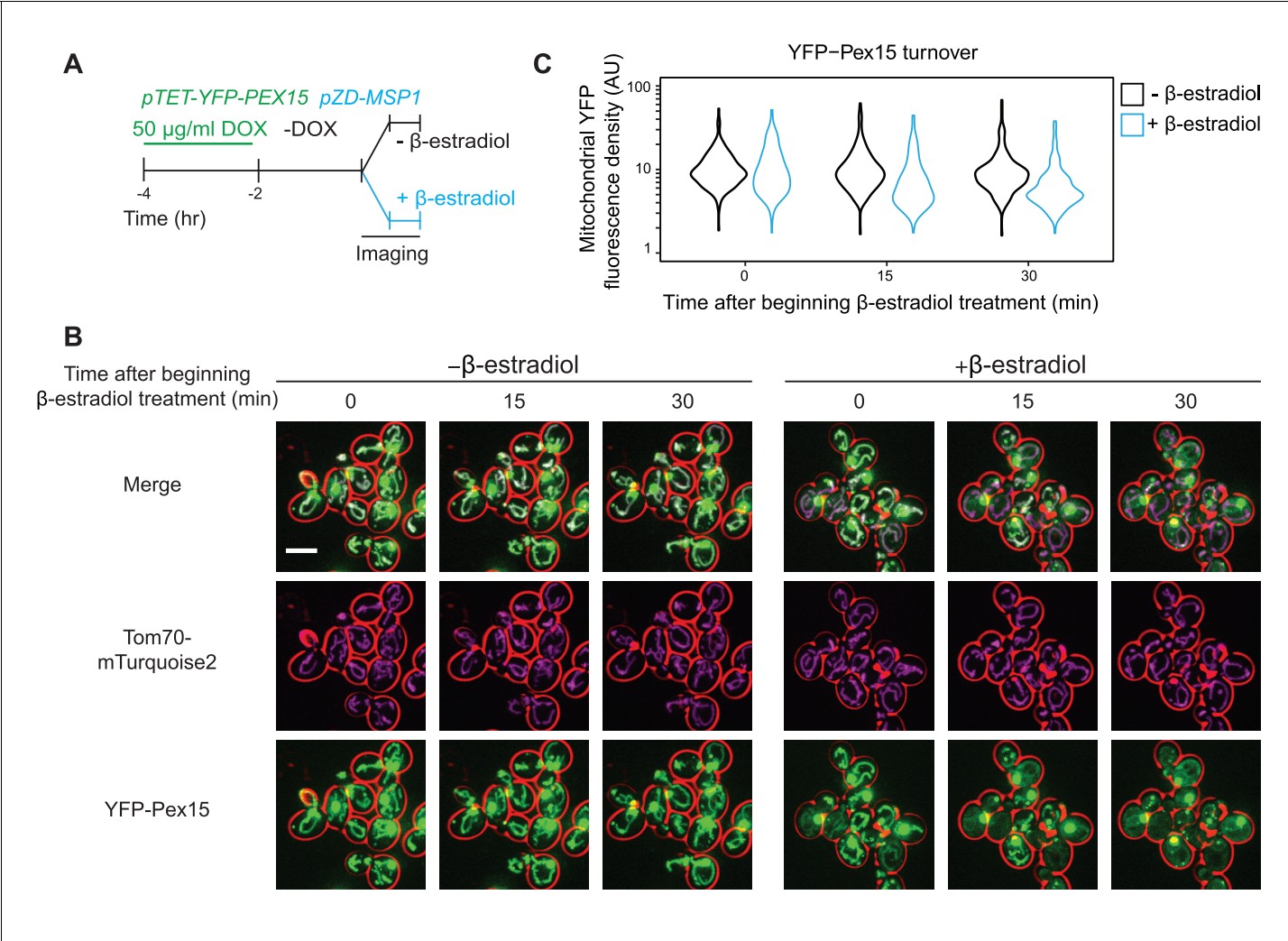

**Figure 1.** Pulse-chase analysis of mitochondrial Pex15 turnover by Msp1. (**A**) Cells containing the doxycycline-inducible promoter driving YFP-Pex15 expression and the β-estradiol-inducible promoter driving Msp1 expression were grown for 2 hr in the presence of 50 µg/ml doxycycline (DOX) before they were washed and grown for 2 hr in drug-free media. *PEX15* mRNAs have a half-life of ~31 min (*Geisberg et al., 2014*), arguing that approximately 7.3% of *PEX15* mRNAs remained when imaging began. This calculation likely overestimates the persistence of *PEX15* mRNA on a per cell basis because it doesn't account for *PEX15* mRNA dilution due to cell division. Following this period of substrate pre-loading, half of the cells were exposed to 1 µM β-estradiol while the other half received vehicle, followed by time-lapse imaging of both cell populations using a spinning disk confocal microscope. This experiment was performed twice with similar results. (**B**) Representative confocal micrographs from the experiment described in part A. Each image represents a maximum intensity projection of a Z-stack. Red cell outlines originate from a single bright-field image acquired at the center of the Z-stack. Scale bar, 5 µm. (**C**) Quantitation of mitochondrial YFP-Pex15 fluorescence from the experiment described in part A. YFP-Pex15 fluorescence density corresponds to the total YFP-Pex15 signal at each computationally-defined mitochondrion (marked by Tom70-mTurquoise2) divided by the mitochondrial pixel volume (see Materials and methods and *Figure 1—figure supplement 2* for more details). Shown are violin plots of the resulting YFP-Pex15 density distributions. These data represent analysis of 123 mock-treated and 93 β-estradiol-treated cells followed throughout the time course as well as progeny from cell divisions during the experiment.

DOI: https://doi.org/10.7554/eLife.28507.003

The following figure supplements are available for figure 1:

**Figure supplement 1.** Supporting evidence for Msp1 turnover of YFP-Pex15 at mitochondria.
DOI: https://doi.org/10.7554/eLife.28507.004

**Figure supplement 2.** Schematic of the processing pipeline for identifying mitochondria and peroxisomes from fluorescence microscopy images.
DOI: https://doi.org/10.7554/eLife.28507.005

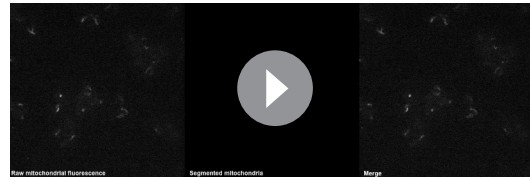

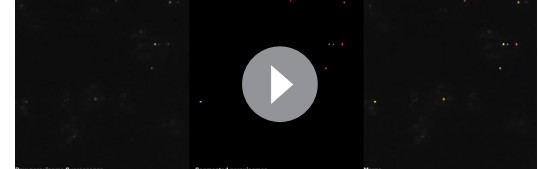

**Video 1.** Representative output from segmenting mitochondria in confocal Z-stacks. An animated Z-stack showing raw Tom70-mTurquoise2 fluorescence visualized by spinning disk confocal microscopy (left), segmented mitochondria identified in the image (middle), and an overlay of the raw image and segmentation output (right). See Materials and methods and *Figure 1—figure supplement 2* for segmentation details. Each contiguous single-color object represents one segmented mitochondrion. Video travels from the bottom to the top of the Z-stack in 0.2 µm slices.

DOI: https://doi.org/10.7554/eLife.28507.008

**Video 2.** Representative output from segmenting peroxisomes in confocal Z-stacks. An animated Z-stack showing raw mCherry-PTS1 fluorescence visualized by spinning disk confocal microscopy (left), segmented peroxisomes identified in the image (middle), and an overlay of the raw image and segmentation output (right). See Materials and methods and *Figure 1—figure supplement 2* for segmentation details. Each contiguous single-color object represents one segmented peroxisome. Video travels from the bottom to the top of the Z-stack in 0.2 µm slices.

DOI: https://doi.org/10.7554/eLife.28507.009

To our knowledge, Msp1-induced turnover of peroxisomal Pex15 had not been reported previously. We found two pieces of evidence that this unexpected phenotype was the product of Pex15 overexpression. First, treatment of *pTET-YFP-PEX15* cells with 5 µg/ml DOX concentration still induced a > 10 fold higher YFP fluorescence at peroxisomes relative to steady state levels of YFP-Pex15 expressed from its native promoter (*Figure 3—figure supplement 1A–B*). Second, we could detect no difference in natively-expressed peroxisomal Pex15 levels when we compared wild-type and *msp1Δ* cells (*Figure 3E*, left panel). This is unlikely a signal detection problem because we could robustly detect the accumulation of natively-expressed Pex15$_{\Delta C30}$ at mitochondria in *msp1Δ* cells (*Figure 3E*, right panel).

Why does Msp1-dependent turnover of peroxisomal Pex15 necessitate excess substrate when the same AAA machine clears mitochondria of even trace amounts of mistargeted Pex15? In search of an answer to this question, we repeated our analysis at higher temporal resolution and found a major difference between the kinetic signatures of mitochondrial and peroxisomal Pex15 turnover by Msp1 (*Figure 4A* and see below). Specifically, while mitochondrial Pex15 turnover showed simple exponential decay (*i.e.* linear decay after log-transformation), the decay of peroxisomal Pex15 appeared to be more complex, comprising faster and slower kinetic components. We detected no major kinetic differences between Msp1 targeting to mitochondria and peroxisomes that could explain this phenomenon (*Figure 1—figure supplement 1B–C*) but found a potential clue from a proteome-wide pulse-chase study showing that while most proteins decay exponentially, some exhibit non-exponential decay that can be explained by their stoichiometric excess over their binding partners (*McShane et al., 2016*). Since peroxisomal membranes have unique residents that interact with native Pex15 (*Eckert and Johnsson, 2003*), we hypothesized that non-exponential decay of overexpressed peroxisomal Pex15 arises due to the existence of an Msp1-sensitive 'solitary' Pex15 state and an Msp1-insensitive 'partner-bound' Pex15 state. This solitary state would be minimally populated by endogenously expressed Pex15 under steady-state conditions, but a significant fraction of overexpressed Pex15 molecules would be solitary because of stoichiometric excess. By contrast, since mitochondria are unlikely to have Pex15-binding partners, mitochondrial Pex15 would exist in an obligate solitary state and would therefore decay exponentially.

To test this hypothesis, we fit our microscopic YFP-Pex15 decay data against two competing stochastic models, which were previously used to describe proteome-wide protein decay data (see Materials and methods for modelling details) (*McShane et al., 2016*). In the 1-state (exponential) model (*Figure 4B*, left), we posit that all Pex15 molecules have the same probability of decay ($k_{decay}$). In the 2-state (non-exponential) model (*Figure 4B*, right), we introduce the probability ($k_{mat}$) of nascent Pex15 maturation, alongside distinct probabilities for decay of the nascent ($k_{decay,1}$) and mature ($k_{decay,2}$) Pex15 states. Depending upon the determined fit parameters, the 2-state model can

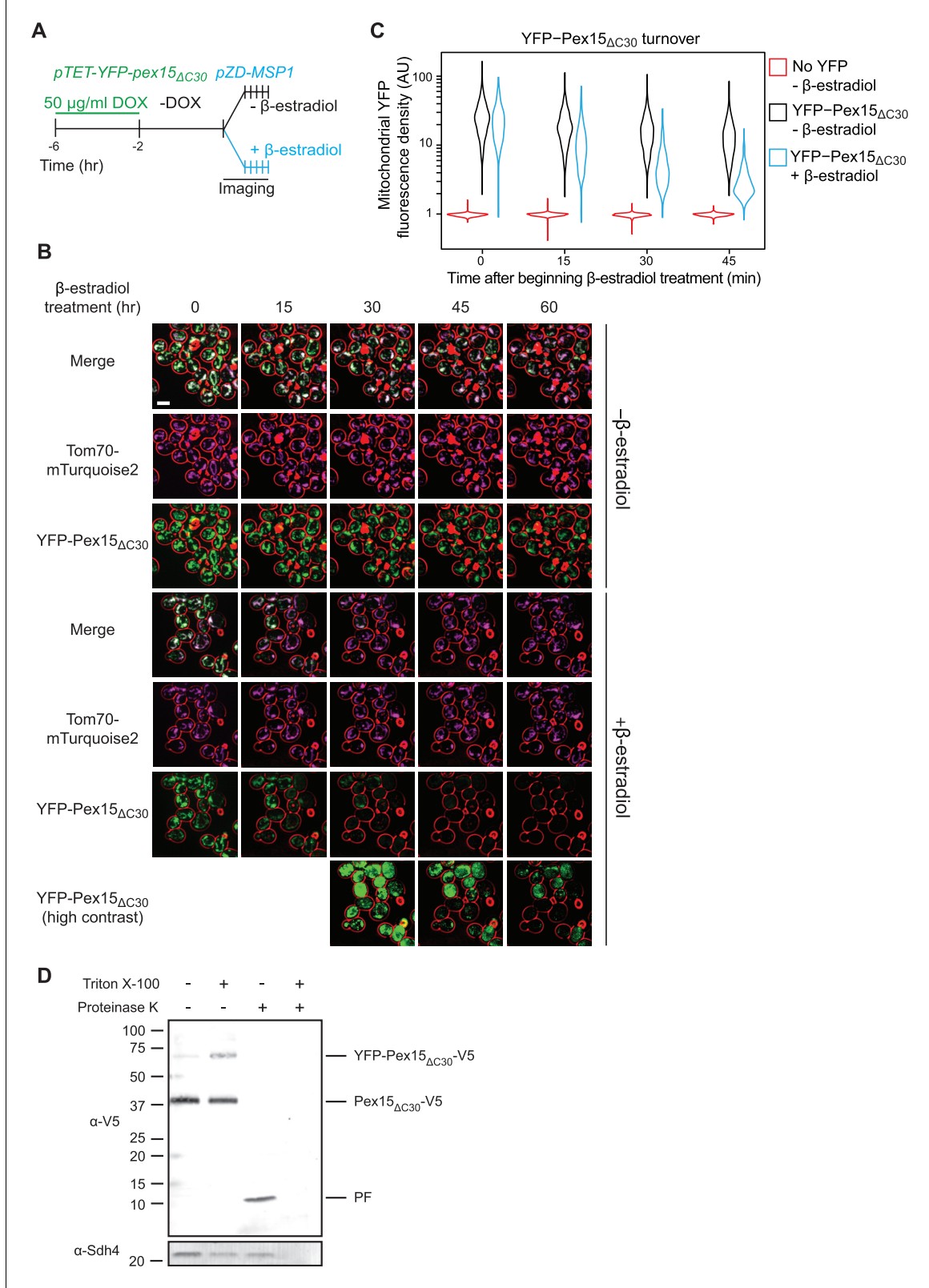

**Figure 2.** Pulse-chase analysis of mitochondrial Pex15$_{\Delta C30}$ turnover by Msp1. (**A**) Experimental timeline of the staged expression experiment for monitoring Msp1-dependent turnover of mitochondrial YFP-Pex15$_{\Delta C30}$. This experiment was performed twice with similar results. (**B**) Representative confocal micrographs from the experiment described in part A. Each image represents a maximum intensity projection of a Z-stack. Red cell outlines originate from a single bright-field image acquired at the center of the Z-stack. Contrast-enhanced YFP-Pex15$_{\Delta C30}$ fluorescence is shown for later

*Figure 2 continued on next page*

*Figure 2 continued*

timepoints +β-estradiol to permit visualization of dim signals. Scale bar, 5 µm. (C) Quantitation of mitochondrial YFP-Pex15$_{\Delta C30}$ fluorescence from the experiment described in part *B*. YFP-Pex15$_{\Delta C30}$ fluorescence density corresponds to the total YFP signal at each computationally-defined mitochondrion (marked by Tom70-mTurquoise2) divided by the mitochondrial pixel volume (see Materials and methods for more details). These data represent analysis of 382 mock-treated and 210 β-estradiol-treated *TET-YFP-PEX15$_{\Delta C30}$* cells and 198 cells lacking YFP-tagged Pex15 followed throughout the time course as well as progeny from cell divisions during the experiment. Laser power was increased from the experiment shown in *Figure 1B–C*, and therefore AUs are not comparable between these experiments. (D) Protease protection assay monitoring YFP-Pex15$_{\Delta C30}$-V5 integration into mitochondria. Crude mitochondria were isolated from *TET-YFP-pex15$_{\Delta C30}$-V5* cells (see Materials and methods for details) and subjected to Proteinase K (PK) or mock treatment in the presence or absence of 1% Triton X-100. Samples were resolved by SDS-PAGE and analyzed by immunoblotting with the indicated antibodies. Immunoblotting with an α-V5 antibody visualized bands at the predicted molecular weight for full-length YFP-Pex15$_{\Delta C30}$-V5 (top), Pex15ΔC30-V5 lacking the YFP tag (middle and *Figure 2—figure supplement 1*), and a smaller protease-resistant fragment (PF, bottom). Immunoblotting was performed against the mitochondrial inner membrane protein Sdh4 to assess accessibility of the mitochondrial intermembrane space to PK. See *Figure 2—figure supplement 1* for α-YFP immunoblotting.

DOI: https://doi.org/10.7554/eLife.28507.006

The following figure supplement is available for figure 2:

**Figure supplement 1.** Supporting evidence for Msp1-dependent turnover of mitochondrial YFP-Pex15$_{\Delta C30}$.

DOI: https://doi.org/10.7554/eLife.28507.007

approximate a 1-state model by minimizing the contribution of one of the two states (*Sin et al., 2016*). To quantify the difference between the 1-state and 2-state models for each sample, and therefore to assess the contribution of a distinct second substrate state to turnover, we measured the area between the 1-state and 2-state fit curves (see Materials and methods).

To analyze mitochondrial Msp1 substrate turnover, we chose YFP-Pex15$_{\Delta C30}$ over wild-type Pex15 to avoid measuring weak mitochondrial signals juxtaposed to strong peroxisomal signals (compare *Figure 1B* and *Figure 2B*). We also restricted our analysis to the first 45 min of β-estradiol treatment because longer Msp1 induction times led to a significant fraction of mitochondria with no detectable YFP signal, which would interfere with turnover fitting (*Figure 2B*, later timepoints). In both the presence and absence of Msp1, our measurements could be similarly explained by both 1-state and 2-state models. The fits from these two models were almost identical (*Figure 4C–D*, *Figure 4G*, and *Figure 4—figure supplement 1A*). Thus, we parsimoniously concluded that Msp1 enhances Pex15 clearance from mitochondria as part of a simple exponential process. Turning to overexpressed YFP-Pex15 at peroxisomes, where YFP-Pex15 persisted at peroxisomes for over 3 hr (*Figure 3B*, later timepoints), we could undertake quantitative analysis on a longer timescale. We again found that the 1-state model and 2-state were indistinguishable in the absence of Msp1. By contrast, the 1-state and 2-state models yielded markedly different fits for our measurements taken after inducing expression of Msp1 (*Figure 4E–G* and *Figure 4—figure supplement 1A–B*). The fit parameters from the 2-state model, which more closely approximated measured Pex15 turnover, revealed that Pex15 in the nascent state decayed ~4 fold faster ($k_{decay,\ 1}$ = 3.45 hr$^{-1}$) than Pex15 in the mature state ($k_{decay,\ 2}$ = 0.87 hr$^{-1}$) (*Figure 4—figure supplement 1A*).

## Msp1 selectively clears newly-resident Pex15 molecules from peroxisomes

The 1-state and 2-state models of peroxisomal Pex15 turnover make distinct predictions about the effect of Msp1 expression on the age of Pex15 molecules. Specifically, in the 1-state model, transient Msp1 overexpression in cells with constitutive Pex15 expression should equally destabilize all Pex15 molecules, thus rapidly reducing their mean age over time (*Figure 5B*, top left panel). By contrast, in the 2-state model, Pex15 age should be buffered against Msp1 overexpression because of two opposing forces (*Figure 4B* and *Figure 5B*, top right panel): At one end, there would be an increase in $k_{decay,1}$ leading to less nascent Pex15, which would drive down the mean age over time. However, there would also be an opposing consequence of rapid depletion of new peroxisomal Pex15 by Msp1: the mature population of Pex15 would receive fewer new (younger) molecules, which would drive up the mean age over time. Notably, both models predict that transient Msp1 expression would result in a decrease in peroxisomal Pex15 levels, albeit with differing kinetics (*Figure 5B*, bottom panels). We simulated Pex15 levels and age following transient Msp1 activation in the 1- and 2-state models with a set of possible half-lives that ranged from our microscopically determined value of 58 min to as slow as

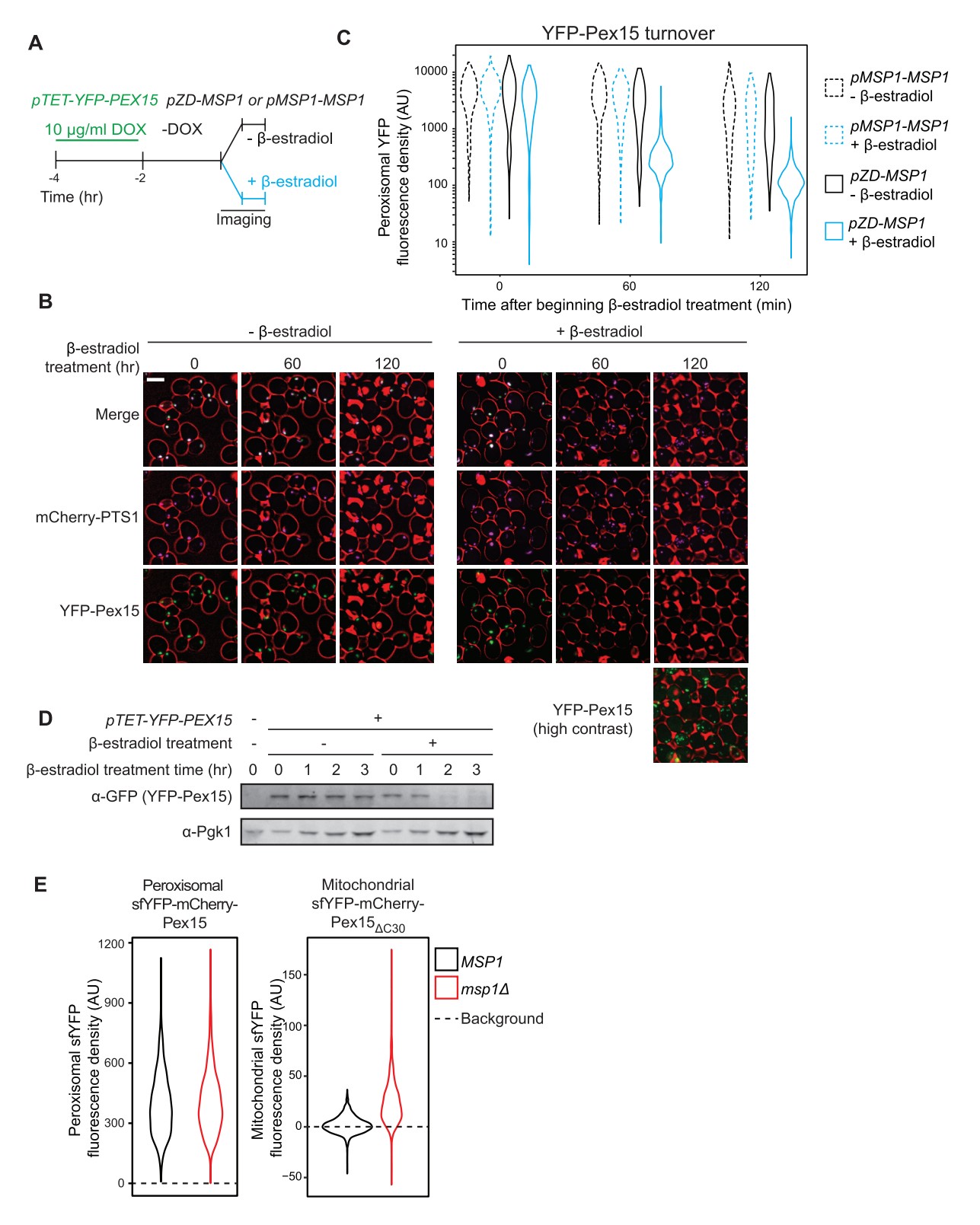

**Figure 3.** Pulse-chase analysis of peroxisomal Pex15 turnover by Msp1. (**A**) Experimental timeline of a pulse-chase analysis similar to the one described in *Figure 1A* but with 10 μg/ml DOX. This experiment was performed twice with similar results. (**B**) Representative confocal micrographs from the experiment described in part *A*. Each image represents a maximum intensity projection of a Z-stack. Red cell outlines originate from a single bright-field image acquired at the center of the Z-stack. Scale bar, 5 μm. Contrast-enhanced YFP-Pex15 fluorescence is shown for the last timepoint +β-

*Figure 3 continued on next page*

*Figure 3 continued*

estradiol to permit visualization of dim signals. (**C**) Quantitation of peroxisomal YFP-Pex15 fluorescence from the experiment described in part A. YFP-Pex15 fluorescence density corresponds to the total YFP-Pex15 signal at each computationally-defined peroxisome (marked by mCherry-PTS1) divided by the peroxisomal pixel volume (see Materials and methods for more details). Shown are violin plots of the resulting YFP-Pex15 density distributions. Solid lines represent cells with Msp1 expression driven by the β-estradiol-inducible *ZD* promoter. Dashed lines represent cells with Msp1 produced from the endogenous *MSP1* promoter. These data represent analysis of 270 mock-treated and 304 β-estradiol-treated *pMSP1-MSP1* cells and 219 mock-treated and 319 β-estradiol-treated *pZD-MSP1* cells followed throughout the time course as well as progeny from cell divisions during the experiment. The 515 nm laser power was decreased relative to the experiments in *Figures 1* and *2* and therefore AUs are not comparable between these experiments. (**D**) Immunoblot analysis of YFP-Pex15 levels after activating *MSP1* expression. Whole cell lysates were prepared from cells grown as described in part A at the indicated timepoints after initiating β-estradiol treatment, and then YFP-Pex15 protein was resolved by SDS-PAGE and immunoblotting. Each sample was prepared from an equal volume of culture to measure turnover of YFP-Pex15 from equivalent amounts of starting material. α-Pgk1 immunoblotting was performed as a loading control. Immunoblotting revealed no significant YFP-Pex15 turnover in the absence of Msp1 induction, whereas the corresponding peroxisomal Pex15 levels dropped somewhat during the timecourse (compare lanes 2–5 to left *Figure 3C* left panels). YFP-Pex15 dilution by cell division may explain this discrepancy. (**E**) Quantitation of endogenously expressed peroxisomal sfYFP-mCherry-Pex15 (left) or mitochondrial sfYFP-mCherry-Pex15$_{\Delta C30}$ (right) sfYFP fluorescence density in wild-type and *msp1Δ* cells. Peroxisomal sfYFP fluorescence density corresponds to the total sfYFP signal at each computationally-defined peroxisome (marked by mTurquoise2-PTS1) divided by the peroxisome volume in pixels. Mitochondrial sfYFP fluorescence density corresponds to the total sfYFP signal at each computationally-defined mitochondrion (marked by Tom70-mTurquoise2) divided by the mitochondrial volume in pixels. Shown are violin plots of the resulting sfYFP fluorescence density distributions. Background represents the mean auto-fluorescence in the sfYFP channel from peroxisomes and mitochondria in strains lacking fluorescently labeled Pex15. Background is normally distributed around the mean and therefore low-fluorescence or non-fluorescent organelles can have negative fluorescence density after background subtraction. These data represent analysis of 941 sfYFP-mCherry-Pex15 *MSP1* cells, 942 sfYFP-mCherry-Pex15 *msp1Δ* cells, 807 sfYFP-mCherry-Pex15$_{\Delta C30}$ *MSP1* cells, and 918 sfYFP-mCherry-Pex15$_{\Delta C30}$ *msp1Δ* cells.
DOI: https://doi.org/10.7554/eLife.28507.010

The following figure supplement is available for figure 3:

**Figure supplement 1.** Supporting evidence that Msp1 induces turnover of overexpressed Pex15 at peroxisomes.
DOI: https://doi.org/10.7554/eLife.28507.011

143 min, as reported in the literature (*Belle et al., 2006*) (*Figure 5B*). Since our half-life value includes decay due to dilution from cell division, it is likely an underestimate of the actual value.

To measure the effect of Msp1 overexpression on the age of Pex15 molecules, we N-terminally tagged natively-expressed Pex15 with a tandem fluorescent timer (tFT-Pex15) (*Figure 5—figure supplement 1A* and *Khmelinskii et al., 2012*) comprising a slow-maturing mCherry and a rapidly-maturing superfolder YFP (sfYFP). On a population level, the mean ratio of mCherry to sfYFP fluorescence is a hyperbolic function of tFT-Pex15 age (*Figure 5—figure supplement 1B* and *Khmelinskii et al., 2012*). In this strain background, we marked peroxisomes using mTurquoise2-PTS1 and induced overexpression of Msp1 from a *ZD* promoter using β-estradiol (*Figure 5A*). Live-cell confocal microscopy combined with computational image analysis revealed a progressive reduction in peroxisomal sfYFP signal following Msp1 overexpression consistent with the predictions of both models, though with kinetics more akin to the predictions of the 2-state model (*Figure 5B–C*, bottom panels). More strikingly, the peroxisomal mCherry:sfYFP fluorescence ratio was insensitive to β-estradiol treatment, consistent with the prediction of the 2-state model (*Figure 5B–C*, top panels). Collectively, our experimental evidence and theoretical analysis strongly support the existence of a Pex15 maturation process at peroxisomes that converts newly-synthesized Pex15 molecules from an Msp1-sensitive to an Msp1-insensitive state.

## Pex3 is a Pex15-interacting protein that protects Pex15 from Msp1-dependent clearance at peroxisomes

To gain insight into the molecular basis of Pex15 maturation at peroxisomes, we hypothesized the existence of peroxisomal proteins that interact with Pex15 and whose absence would reveal that natively-expressed Pex15 is a latent substrate for Msp1. The cytosolic AAA proteins Pex1 and Pex6 are two prime candidates for testing this hypothesis because they form a ternary complex with Pex15 (*Birschmann et al., 2003*). However, we did not observe the expected decrease in YFP-Pex15 levels in *pex1Δ* or *pex6Δ* cells that would be indicative of enhanced turnover by Msp1 (*Figure 6—figure supplement 1A*). To look for additional Pex15 binding partners, we noted that the Pex1/6/15 complex is a regulator of peroxisome destruction by selective autophagy (*Kamber et al., 2015*; *Nuttall et al., 2014*). This process is initiated by Atg36, a receptor protein bound to the peroxisomal

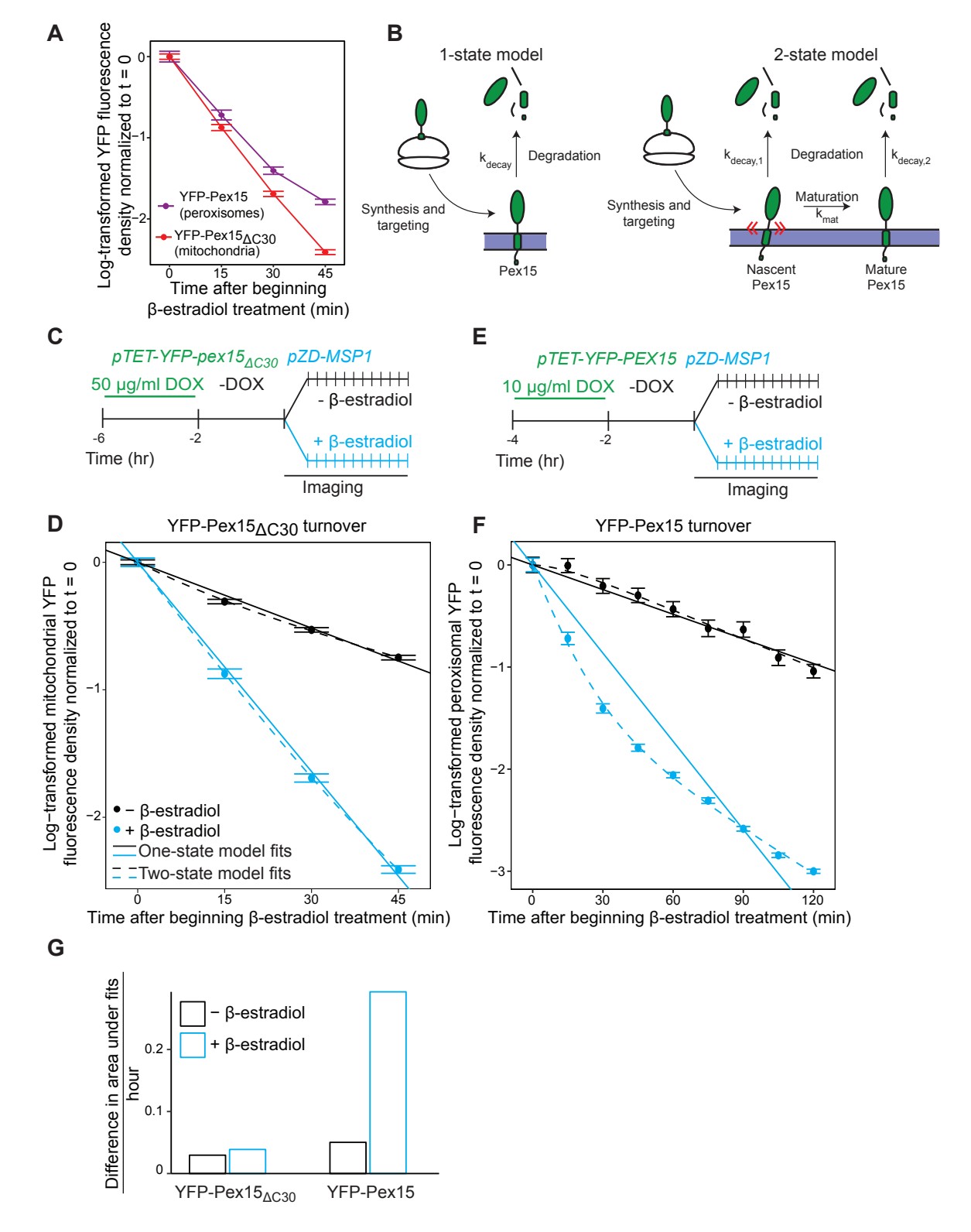

**Figure 4.** Experimental and theoretical evidence for the 2-state model of Pex15 turnover at peroxisomes. (A) The apparent difference in the kinetic profiles of Msp1-induced substrate turnover at peroxisomes (Pex15) versus mitochondria (Pex15$_{\Delta C30}$). These data represent quantitation of data from the experiments described in *Figure 2* (YFP-Pex15$_{\Delta C30}$) and additional timepoints from the experiment described in *Figure 3* (YFP-Pex15). YFP signal density at mitochondria (red) or peroxisomes (purple) is plotted after normalization to the 0 hr timepoint, with lines directly connecting timepoints. Error

*Figure 4 continued on next page*

*Figure 4 continued*

bars represent standard error of the mean. These data are reproduced in parts *D* and *F*. (B) Schematics of the two competing models for Pex15 turnover. In the 1-state model, newly-synthesized Pex15 is first targeted and inserted into the peroxisome membrane and then degraded by a simple exponential decay process that occurs with the rate constant $k_{decay}$. In the 2-state model, there is an additional exponential maturation process that converts Pex15 from a nascent state to a mature state at a rate defined by $k_{mat}$. In addition, this model includes the new exponential decay constant $k_{decay,2}$ for the mature Pex15 state that is distinct from the $k_{decay,1}$ of the nascent state. (C) Experimental timeline of the staged expression experiment for monitoring Msp1-dependent turnover of mitochondrial Pex15$_{\Delta C30}$ with high temporal resolution. (D) Quantitation of mitochondrial YFP-Pex15$_{\Delta C30}$ fluorescence from the experiment described in part *C*. YFP signal density at mitochondria was determined as described in *Figure 1* and plotted after normalization to the 0 hr timepoint. Error bars represent standard error of the mean. Data were fitted to the competing models described in part *B* as indicated (See Materials and methods for model fitting details). See *Figure 4—figure supplement 1A* for fit parameters. (E) Experimental timeline of the staged expression experiment for monitoring Msp1-dependent turnover of peroxisomal Pex15 with high temporal resolution. This experiment was performed twice with similar results. (F) Quantitation of peroxisomal YFP-Pex15 fluorescence from the experiment described in part *E*. YFP signal density at peroxisomes was determined as described in *Figure 3C* and plotted as in part *D*. See *Figure 4—figure supplement 1A* for fit parameters. See *Figure 4—figure supplement 1B* for a similar plot containing only 0–45 min timepoints as plotted for YFP-Pex15$_{\Delta C30}$ in part *D*. (G) Area between the 1-state and 2-state fits shown in parts *D* and *F*. See Materials and methods for details. Total area between curves is divided by time to normalize between fits from different time scales.

DOI: https://doi.org/10.7554/eLife.28507.012

The following figure supplement is available for figure 4:

**Figure supplement 1.** Supporting evidence for the 2-state model of Pex15 turnover at peroxisomes.
DOI: https://doi.org/10.7554/eLife.28507.013

membrane protein Pex3 (*Motley et al., 2012*). Consistent with a previously published split-ubiquitin assay for detecting protein-protein interactions (*Eckert and Johnsson, 2003*), we found that Pex15 interacts with Pex3 by co-immunoprecipitation analysis (*Figure 6A*). Before we could test if Pex3 protects Pex15 from Msp1-dependent turnover, we had to overcome a major technical challenge. Specifically, Pex3 is essential for targeting of numerous peroxisomal membrane proteins, which is why *pex3Δ* cells lack functional peroxisomes (*Fang et al., 2004*). Since Pex3 is normally turned over very slowly (*Figure 6—figure supplement 1D* and *Belle et al., 2006*), promoter shut-off is not a suitable method for acutely depleting Pex3. Instead, we exploited an established Auxin-inducible degradation system to rapidly eliminate Pex3 from peroxisomes in situ. First, we appended a tandem V5 epitope tag followed by an Auxin-inducible degron sequence (*Nishimura et al., 2009*) to the cytosolic C-terminus of Pex3 (Pex3-V5-AID). Next, we overexpressed an E3 ubiquitin ligase from rice (OsTir1) that binds and ubiquitinates Auxin-bound AID to enable degradation of AID fusions by the proteasome (*Nishimura et al., 2009*). Immunoblotting analysis for the V5 epitope revealed that Auxin addition induced rapid Pex3 destruction, which was dependent on OsTir1 expression and independent of Msp1 (*Figure 6—figure supplement 1B–E*). Importantly, microscopic analysis of cells co-expressing Pex3-GFP-AID and mCherry-PTS1 revealed that peroxisomes persisted for hours following Pex3 destruction (*Figure 6—figure supplement 1B*).

We next introduced the Pex3 AID system into either wild-type or *msp1Δ* cells with endogenously expressed tFT-Pex15. To monitor changes in peroxisomal sfYFP fluorescence density after Pex3 depletion we again used live-cell confocal microscopy combined with computational image analysis (*Figure 6B*). Strikingly, we observed that Pex3 degradation immediately increased the rate of Msp1-dependent Pex15 turnover (*Figure 6C*), thus unmasking endogenous Pex15 as a latent substrate. By contrast, Pex3 degradation did not result in Msp1-dependent destabilization of Pex11 and Pex12, two peroxisomal membrane proteins we analyzed as controls for the substrate specificity of Msp1 (*Figure 6—figure supplement 1I–J*). We observed a similar phenomenon in cells overexpressing YFP-Pex15, albeit to a lesser extent, possibly because of excess YFP-Pex15 relative to endogenous Pex3 prior to Auxin addition (*Figure 6—figure supplement 1F–H*). Consistent with this idea, constitutive overexpression of Pex3 from the strong *TDH3* promoter blunted the effect of de novo Msp1 induction on transiently overexpressed YFP-Pex15 (*Figure 6D–E*). Taken together, these data argue that Pex3 stoichiometrically protects Pex15 from Msp1 recognition at peroxisomes.

## Organelle-restricted Pex15 clearance by Msp1 with artificial transmembrane anchors

A recent study showed that GFP fused to the TMS of the mammalian Msp1 homolog ATAD1 is targeted to both mitochondria and peroxisomes (*Liu et al., 2016*). This suggests that the TMS of Msp1 is

an ambiguous targeting signal whose function is to localize the rest of Msp1 into proximity with its substrates. To explore this issue, we first attempted to restrict Msp1 to either mitochondria or peroxisomes by replacing Msp1's TMS with the signal anchor of Tom70 (Tom70$_{TMS}$-Msp1), a mitochondrial outer membrane resident, or the transmembrane peroxisomal targeting signal of Pex22 (Pex22$_{TMS}$-Msp1), respectively (*Figure 7A*). Indeed, Tom70$_{TMS}$-Msp1-YFP produced from the *MSP1* promoter is primarily localized to mitochondria with some residual localization to peroxisomes, whereas Pex22$_{TMS}$-Msp1-YFP was exclusively localized to peroxisomes (*Figure 7B* and *Figure 7—figure supplement 1A*). Next, we monitored the ability of these Msp1 chimeras to suppress mitochondrial accumulation of tFT-Pex15$_{\Delta C30}$ in cells lacking wild type Msp1 and found that Tom70$_{TMS}$-Msp1 was fully functional, whereas Pex22$_{TMS}$-Msp1 was unable to complement the *msp1Δ* phenotype (*Figure 7C* and *Figure 7—figure supplement 1B*). Lastly, we monitored clearance of excess peroxisomal YFP-Pex15 following de novo induction of Msp1 chimaeras (*Figure 7D*). This analysis revealed that Pex22$_{TMS}$-Msp1 enhanced substrate turnover more robustly than Tom70$_{TMS}$-Msp1 (*Figure 7E*), which we can simply explain by its relatively higher peroxisome abundance (*Figure 7B*). These data lead us to speculate that the Msp1 AAA domain (with its juxtamembrane region) initiates substrate clearance by directly binding to substrate regions at the interface between the aqueous cytosol and the lipid core.

## Discussion

Errors in TA protein targeting by the GET pathway pose a constant threat to mitochondrial health. Two recent studies revealed that yeast Msp1 (ATAD1 in humans), a AAA membrane protein resident on the surface of mitochondria and peroxisomes, is part of a conserved mechanism for preventing mistargeted TA proteins from accumulating in mitochondria (*Chen et al., 2014b*; *Okreglak and Walter, 2014*). At the same time, this pioneering work raised an important question about Msp1's substrate selectivity: What distinguishes TA proteins mistargeted to mitochondria from TA proteins native to mitochondria and peroxisomes?

Here, we answer this question as it pertains to Pex15, a native peroxisomal TA protein known to be an Msp1 substrate when mistargeted to mitochondria (*Chen et al., 2014b*; *Okreglak and Walter, 2014*). As our starting point, we coupled live-cell quantitative microscopy with two orthogonal drug-inducible gene-expression systems to show that de novo induction of Msp1 activity clears a fully-integrated Pex15 variant from mitochondria (*Figure 7*). This result solidifies the working model in the literature that Msp1 is a mechanoenzyme capable of extracting its substrates from the membrane (*Chen et al., 2014b*; *Okreglak and Walter, 2014*; *Wohlever et al., 2017*). We were also able to reveal that peroxisomal Pex15 is a latent Msp1 substrate at peroxisomes. The key starting observation that led us to this conclusion was that Pex15 overexpressed at peroxisomes was turned over by an unusual non-exponential process, which depended on Msp1 induction. By model fitting of these data and comparative analysis with the exponential decay of mitochondrial Pex15, we found evidence for a Pex15 maturation mechanism unique to peroxisomes. By positing that this mechanism converts newly-resident peroxisomal Pex15 from an initial Msp1-sensitive state to an Msp1-resistant state, we were able to account for the non-exponential decay kinetics (*Figure 8*). Moreover, we validated a key prediction of this mechanism by showing that Msp1 selectively removes peroxisomal Pex15 from the young end of its molecular age distribution. More broadly, a testable hypothesis that emerges as an extension of our work is that native mitochondrial TA proteins are latent substrates normally shielded from Msp1 by maturation mechanisms specific to mitochondria.

The precise molecular mechanism by which Pex15 matures into an Msp1-resistant state remains to be worked out. However, our evidence strongly argues that complex assembly between Pex15 and the peroxisomal membrane protein Pex3 is a critical component of this process. Pex3 has been previously shown to play a role in the insertion of peroxisomal membrane proteins (*Fang et al., 2004*). Thus, it is possible that loss of Pex3 function leads to indirect loss of another membrane protein that itself blocks Msp1-dependent turnover of Pex15. We cannot formally exclude this possibility but we find it unlikely for three reasons. First, we showed that Pex3 co-immunoprecipitates with Pex15. Thus, in principle, Pex3 could physically occlude an Msp1 binding site on Pex15 or make Pex15 structurally more resistant to mechanodisruption. Second, we showed that rapid degradation of Pex3 causes a near-instantaneous increase in the rate of Msp1-dependent Pex15 clearance from peroxisomes without destabilizing two control peroxisomal membrane proteins. Third, we found that overproduction of Pex3 increased protection of overexpressed Pex15 from Msp1-dependent

turnover at peroxisomes. Our results do not rule out the possibility that additional binding partners of Pex15, such as certain components of the importomer for peroxisomal matrix proteins (*Rosenkranz et al., 2006*), confer protection from Msp1. More broadly, a simple extension of our working model for Msp1 substrate selectivity leads to the intriguing hypothesis that native mitochondrial TA proteins are shielded from Msp1 by their binding partners. The microscopy methodology we have described here will facilitate testing of this idea in the near future.

Lastly, our work adds Msp1 to the growing class of proteostasis pathways that mediate degradation of excess subunits of soluble (*Sung et al., 2016*) and transmembrane complexes (*Kihara et al., 1995*; *Lippincott-Schwartz et al., 1988*; *Westphal et al., 2012*). Interestingly, Msp1 is expressed at a relatively low level (*Ghaemmaghami et al., 2003*) and its prolonged overexpression induces severe growth defects (data not shown). This raises the possibility that superphysiological levels of Msp1 are detrimental because they reduce the abundance of undefined protein complexes via hypervigilant membrane clearance of immature subunits and complex assembly intermediates. Future tests of this idea using proteome-wide approaches have the potential to define the full breadth of Msp1's role in maintaining protein complex homeostasis.

# Materials and methods

## Key resources table

| Reagent type (species) or resource | Designation | Source or reference | Identifiers | Additional information |
|---|---|---|---|---|
| strain, strain background (*S. cerevisiae*) | BY4741 | PMID: 9483801 | | |
| strain, strain background (*S. cerevisiae*) | BY4741 trp1Δ::pTDH3-mTURQUOISE2-PTS1-SpHIS5 | This paper | Euroscarf:VDY3349 | |
| strain, strain background (*S. cerevisiae*) | BY4741 trp1Δ::pTDH3-mTURQUOISE2-PTS1-SpHIS5 ura3Δ::pPEX15-sfYFP-mCHERRY-PEX15-tPEX15-KANMX | This paper | Euroscarf:VDY3350 | |
| strain, strain background (*S. cerevisiae*) | BY4741 trp1Δ::pTDH3-mTURQUOISE2-PTS1-SpHIS5 ura3Δ::pPEX15-sfYFP-mCHERRY-PEX15-tPEX15-KANMX msp1Δ::HPHMX | This paper | Euroscarf:VDY3351 | |
| strain, strain background (*S. cerevisiae*) | BY4741 trp1Δ::pTDH3-mTURQUOISE2-PTS1-SpHIS5 ura3Δ::pPEX15-sfYFP-mCHERRY-PEX15-tPEX15-KANMX CgLEU2-Z4EV-pZD-MSP1 | This paper | Euroscarf:VDY3352 | |
| strain, strain background (*S. cerevisiae*) | BY4741 trp1Δ::pTDH3-mTURQUOISE2-PTS1-SpHIS5 ura3Δ::pPEX15-sfYFP-mCHERRY-PEX15-tPEX15-KANMX pex15Δ::URA | This paper | Euroscarf:VDY3516 | |
| strain, strain background (*S. cerevisiae*) | BY4741 trp1Δ::pTDH3-mTURQUOISE2-PTS1-SpHIS5 ura3Δ::pPEX15-sfYFP-mCHERRY-PEX15-tPEX15-KANMX pex15Δ::URA msp1Δ::HPHMX PEX3-3FLAG-NATMX | This paper | Euroscarf:VDY3518 | |
| strain, strain background (*S. cerevisiae*) | BY4741 trp1Δ::pTDH3-mTURQUOISE2-PTS1-SpHIS5 pex15Δ::URA msp1Δ::HPHMX PEX3-3FLAG-NATMX | This paper | Euroscarf:VDY3519 | |
| strain, strain background (*S. cerevisiae*) | BY4741 TOM70-mTURQUOISE2-SpHIS5 | This paper | Euroscarf:VDY3354 | |

| | | | |
|---|---|---|---|
| strain, strain background (*S. cerevisiae*) | *BY4741 TOM70-mTURQUOISE2-SpHIS5 ura3Δ::pPEX15-sfYFP-mCHERRY-pex15$_{ΔC30}$-tPEX15-KANMX* | This paper | Euroscarf:VDY3355 |
| strain, strain background (*S. cerevisiae*) | *BY4741 TOM70-mTURQUOISE2-SpHIS5 ura3Δ::pPEX15-sfYFP-mCHERRY-pex15$_{ΔC30}$-tPEX15-KANMX msp1Δ::HPHMX* | This paper | Euroscarf:VDY3356 |
| strain, strain background (*S. cerevisiae*) | *BY4741 TOM70-mTURQUOISE2-SpHIS5 trp1Δ::pTDH3-mCHERRY-PTS1-CgURA3* | This paper | Euroscarf:VDY3357 |
| strain, strain background (*S. cerevisiae*) | *BY4741 TOM70-mTURQUOISE2-SpHIS5 trp1Δ::pTDH3-mCHERRY-PTS1-CgURA3 MSP1-YFP-KANMX* | This paper | Euroscarf:VDY3358 |
| strain, strain background (*S. cerevisiae*) | *BY4741 TOM70-mTURQUOISE2-SpHIS5 trp1Δ::pTDH3-mCHERRY-PTS1-CgURA3 CgLEU2-Z4EV-pZD-MSP1-YFP-KANMX* | This paper | Euroscarf:VDY3359 |
| strain, strain background (*S. cerevisiae*) | *BY4741 TOM70-mTURQUOISE2-SpHIS5 trp1Δ::pTDH3-mCHERRY-PTS1-CgURA3 CgLEU2-Z4EV-pZD-MSP1* | This paper | Euroscarf:VDY3360 |
| strain, strain background (*S. cerevisiae*) | *BY4741 trp1Δ::pTDH3-mCHERRY-PTS1-CgURA3 YFP-PEX15-tPEX15-KANMX* | This paper | Euroscarf:VDY3001 |
| strain, strain background (*S. cerevisiae*) | *BY4741 TOM70-mTURQUOISE2-SpHIS5 trp1Δ::pTDH3-mCHERRY-PTS1-CgURA3 ura3Δ::CgTRP1-rTA-pTET-YFP-PEX15-tPEX15* | This paper | Euroscarf:VDY3607 |
| strain, strain background (*S. cerevisiae*) | *BY4741 TOM70-mTURQUOISE2-SpHIS5 trp1Δ::pTDH3-mCHERRY-PTS1-CgURA3 CgLEU2-Z4EV-pZD-MSP1 ura3Δ::CgTRP1-rTA-pTET-YFP-PEX15-tPEX15* | This paper | Euroscarf:VDY3527 |
| strain, strain background (*S. cerevisiae*) | *BY4741 TOM70-mTURQUOISE2-SpHIS5 trp1Δ::pTDH3-mCHERRY-PTS1-CgURA3 CgLEU2-Z4EV-pZD-MSP1 ura3Δ::CgTRP1-rTA-pTET-YFP-PEX15-tPEX15 NATMX-pTDH3-PEX3* | This paper | Euroscarf:VDY3608 |
| strain, strain background (*S. cerevisiae*) | *BY4741 TOM70-mTURQUOISE2-SpHIS5 trp1Δ::pTDH3-mCHERRY-PTS1-CgURA3 CgLEU2-Z4EV-pZD-MSP1 ura3Δ::CgTRP1-rTA-pTET-YFP-pex15$_{ΔC30}$-tPEX15* | This paper | Euroscarf:VDY3362 |
| strain, strain background (*S. cerevisiae*) | *BY4741 TOM70-mTURQUOISE2-SpHIS5 trp1Δ::pTDH3-mCHERRY-PTS1-CgURA3 CgLEU2-Z4EV-pZD-MSP1 ura3Δ::CgTRP1-rTA-pTET-YFP-pex15$_{ΔC30}$-V5-tPEX15* | This paper | Euroscarf:VDY3412 |

| strain, strain background (S. cerevisiae) | BY4741 trp1Δ::pTDH3-mCherry-PTS1::HPHMX, Pex3-GFP-AID-HIS3M × 6 leu2Δ::pTDH3-OsTIR1-CgLEU2 | This paper | Euroscarf:VDY2837 |
|---|---|---|---|
| strain, strain background (S. cerevisiae) | BY4741 PEX3-V5-AID-KANMX | This paper | Euroscarf:VDY2770 |
| strain, strain background (S. cerevisiae) | BY4741 PEX3-V5-AID-KANMX leu2Δ::pTDH3-OsTIR1-CgLEU2 | This paper | Euroscarf:VDY2773 |
| strain, strain background (S. cerevisiae) | BY4741 PEX3-V5-AID-KANMX leu2Δ::pTDH3-OsTIR1-CgLEU2 msp1Δ::HIS | This paper | Euroscarf:VDY3399 |
| strain, strain background (S. cerevisiae) | BY4741 TOM70-mTURQUOISE2-SpHIS5 trp1Δ::pTDH3-mCHERRY-PTS1-CgURA3 ura3Δ::CgTRP1-rTA-pTET-YFP-PEX15-tPEX15 PEX3-V5-AID-KANMX leu2Δ::pTDH3-OsTIR1-CgLEU2 | This paper | Euroscarf:VDY3363 |
| strain, strain background (S. cerevisiae) | BY4741 TOM70-mTURQUOISE2-SpHIS5 trp1Δ::pTDH3-mCHERRY-PTS1-CgURA3 ura3Δ::CgTRP1-rTA-pTET-YFP-PEX15-tPEX15 PEX3-V5-AID-KANMX leu2Δ::pTDH3-OsTIR1-CgLEU2 msp1Δ::HPHMX | This paper | Euroscarf:VDY3364 |
| strain, strain background (S. cerevisiae) | BY4741 PEX11-mTURQUOISE2-SpHIS5 | This paper | Euroscarf:VDY3444 |
| strain, strain background (S. cerevisiae) | BY4741 PEX11-mTURQUOISE2-SpHIS5 ura3Δ::pPEX15-sfYFP-mCHERRY-PEX15-tPEX15-KANMX | This paper | Euroscarf:VDY3445 |
| strain, strain background (S. cerevisiae) | BY4741 PEX11-mTURQUOISE2-SpHIS5 ura3Δ::pPEX15-sfYFP-mCHERRY-PEX15-tPEX15-KANMX pex1Δ::NATMX | This paper | Euroscarf:VDY3446 |
| strain, strain background (S. cerevisiae) | BY4741 PEX11-mTURQUOISE2-SpHIS5 ura3Δ::pPEX15-sfYFP-mCHERRY-PEX15-tPEX15-KANMX pex6Δ::URA | This paper | Euroscarf:VDY3447 |
| strain, strain background (S. cerevisiae) | BY4741 PEX3-V5-AID-KANMX trp1Δ::pTDH3-mTURQUOISE2-PTS1-SpHIS5 | This paper | Euroscarf:VDY3528 |
| strain, strain background (S. cerevisiae) | BY4741 PEX3-V5-AID-KANMX trp1Δ::pTDH3-mTURQUOISE2-PTS1-SpHIS5 leu2Δ::pTDH3-OsTIR1-CgLEU2 | This paper | Euroscarf:VDY3529 |
| strain, strain background (S. cerevisiae) | BY4741 PEX3-V5-AID-KANMX trp1Δ::pTDH3-mTURQUOISE2-PTS1-SpHIS5 PEX11-sfYFP-mCHERRY-CgURA3 leu2Δ::pTDH3-OsTIR1-CgLEU2 | This paper | Euroscarf:VDY3609 |
| strain, strain background (S. cerevisiae) | BY4741 PEX3-V5-AID-KANMX trp1Δ::pTDH3-mTURQUOISE2-PTS1-SpHIS5 PEX11-sfYFP-mCHERRY-CgURA3 leu2Δ::pTDH3-OsTIR1-CgLEU2 msp1Δ::HPHMX | This paper | Euroscarf:VDY3610 |

| strain, strain background (S. cerevisiae) | BY4741 PEX3-V5-AID-KANMX trp1Δ::pTDH3-mTURQUOISE2-PTS1-SpHIS5 PEX12-sfYFP-mCHERRY-CgURA3 leu2Δ::pTDH3-OsTIR1-CgLEU2 | This paper | Euroscarf:VDY3611 |
|---|---|---|---|
| strain, strain background (S. cerevisiae) | BY4741 PEX3-V5-AID-KANMX trp1Δ::pTDH3-mTURQUOISE2-PTS1-SpHIS5 PEX12-sfYFP-mCHERRY-CgURA3 leu2Δ::pTDH3-OsTIR1-CgLEU2 msp1Δ::HPHMX | This paper | Euroscarf:VDY3612 |
| strain, strain background (S. cerevisiae) | BY4741 PEX3-V5-AID-URA trp1Δ::pTDH3-mTURQUOISE2-PTS1-SpHIS5 ura3Δ::pPEX15-sfYFP-mCHERRY-PEX15-tPEX15-KANMX | This paper | Euroscarf:VDY3615 |
| strain, strain background (S. cerevisiae) | BY4741 PEX3-V5-AID-URA trp1Δ::pTDH3-mTURQUOISE2-PTS1-SpHIS5 ura3Δ::pPEX15-sfYFP-mCHERRY-PEX15-tPEX15-KANMX msp1Δ::HPHMX | This paper | Euroscarf:VDY3616 |
| strain, strain background (S. cerevisiae) | BY4741 PEX3-V5-AID-URA trp1Δ::pTDH3-mTURQUOISE2-PTS1-SpHIS5 ura3Δ::pPEX15-sfYFP-mCHERRY-PEX15-tPEX15-KANMX leu2Δ::pTDH3-OsTIR1-CgLEU2 | This paper | Euroscarf:VDY3613 |
| strain, strain background (S. cerevisiae) | BY4741 PEX3-V5-AID-URA trp1Δ::pTDH3-mTURQUOISE2-PTS1-SpHIS5 ura3Δ::pPEX15-sfYFP-mCHERRY-PEX15-tPEX15-KANMX leu2Δ::pTDH3-OsTIR1-CgLEU2 msp1Δ::HPHMX | This paper | Euroscarf:VDY3614 |
| strain, strain background (S. cerevisiae) | BY4741 TOM70-mTURQUOISE2-SpHIS5 trp1Δ::pTDH3-mCHERRY-PTS1-CgURA3 PEX22$_{TMS}$-MSP1-YFP | This paper | Euroscarf:VDY3617 |
| strain, strain background (S. cerevisiae) | BY4741 TOM70-mTURQUOISE2-SpHIS5 trp1Δ::pTDH3-mCHERRY-PTS1-CgURA3 TOM70$_{TMS}$-MSP1-YFP | This paper | Euroscarf:VDY3618 |
| strain, strain background (S. cerevisiae) | BY4741 TOM70-mTURQUOISE2-SpHIS5 ura3Δ::pPEX15-sfYFP-mCHERRY-pex15$_{ΔC30}$-tPEX15-KANMX PEX22$_{TMS}$-MSP1 | This paper | Euroscarf:VDY3619 |
| strain, strain background (S. cerevisiae) | BY4741 TOM70-mTURQUOISE2-SpHIS5 ura3Δ::pPEX15-sfYFP-mCHERRY-pex15$_{ΔC30}$-tPEX15-KANMX TOM70$_{TMS}$-MSP1 | This paper | Euroscarf:VDY3620 |
| strain, strain background (S. cerevisiae) | BY4741 TOM70-mTURQUOISE2-SpHIS5 trp1Δ::pTDH3-mCHERRY-PTS1-CgURA3 CgLEU2-Z4EV-pZD-PEX22$_{TMS}$-MSP1 ura3Δ::CgTRP1-rTA-pTET-YFP-PEX15-tPEX15 | This paper | Euroscarf:VDY3621 |

| | | | |
|---|---|---|---|
| strain, strain background (*S. cerevisiae*) | *BY4741 TOM70-mTURQUOISE2-SpHIS5 trp1Δ::pTDH3-mCHERRY-PTS1-CgURA3 CgLEU2-Z4EV-pZD-TOM70_{TMS}-MSP1 ura3Δ::CgTRP1-rTA-pTET-YFP-PEX15-tPEX15* | This paper | Euroscarf:VDY3622 |
| antibody | Mouse anti-V5 monoclonal | Invitrogen | Cat #R960 |
| antibody | Mouse anti-FLAG monoclonal (M2) | Sigma | Cat #F3165 |
| antibody | Mouse anti-mCherry monoclonal (1C51) | Novus | Cat #NBP1-96752 |
| antibody | Mouse anti-Pgk1 monoclonal | Thermo Fisher | Cat #459250 |
| antibody | Rabbit anti-Hsc82 polyclonal | Abcam | Cat #ab30920 |
| antibody | Rabbit anti-Sdh4 polyclonal | Pfanner Lab | Kind gift of N. Pfanner |
| antibody | Mouse anti-GFP monoclonal | Sigma | Cat # 11814460001 |
| antibody | Goat anti-mouse IgG (H + L)-HRP conjugated secondary | Bio-Rad | Cat #170–6515 |
| recombinant DNA reagent | pKT211 (pKT-YFP-SpHIS5) (plasmid) | PMID: 15197731 | |
| recombinant DNA reagent | pKT-mTURQUOISE2-SpHIS5 (plasmid) | This paper | |
| recombinant DNA reagent | pKT-pTDH3-mTURQUOISE2-SpHIS5 (plasmid) | This paper | |
| recombinant DNA reagent | pKT-pTDH3-mCHERRY-CgURA3 (plasmid) | This paper | |
| recombinant DNA reagent | pNH604-rTA-pTET-YFP-PEX15 (plasmid) | This paper | |
| recombinant DNA reagent | pNH604-rTA-pTET-YFP-pex15ΔC30 (plasmid) | This paper | |
| recombinant DNA reagent | pFA6a-pPEX15-sfYFP-mCHERRY-PEX15-KANMX (plasmid) | This paper | |
| recombinant DNA reagent | pFA6a-pPEX15-sfYFP-mCHERRY-pex15ΔC30-KANMX (plasmid) | This paper | |
| recombinant DNA reagent | pFA6a-V5-AID-KANMX (plasmid) | PMID: 27798238 | |
| recombinant DNA reagent | pFA6a-V5-AID-URA3 (plasmid) | PMID: 27798238 | |
| recombinant DNA reagent | pFA6a-GFP-AID-HIS3M × 6 (plasmid) | PMID: 27798238 | |
| recombinant DNA reagent | pNH605-OsTIR1 (plasmid) | PMID: 27798238 | |
| recombinant DNA reagent | pNH605-Z4EV-pZD (plasmid) | This study | |
| recombinant DNA reagent | p3FLAG-NATMX (plasmid) | PMID: 17719544 | |
| peptide, recombinant protein | Proteinase K | Sigma | Cat #3115879001 |
| commercial assay or kit | SuperSignal West Femto Substrate | Thermo Fisher | Cat #34095 |

| | | | |
|---|---|---|---|
| chemical compound, drug | 3-indoleacetic acid (Auxin) | Sigma | Cat #I3750 |
| chemical compound, drug | β-Estradiol | Sigma | Cat #E8875 |
| chemical compound, drug | Doxycycline | Sigma | Cat #D3447 |
| chemical compound, drug | Cycloheximide | Sigma | Cat #C7698 |
| chemical compound, drug | Concanavalin A | MP Biomedicals | Cat #2195283 |
| chemical compound, drug | 3 × FLAG peptide | Sigma | Cat #F4799 |
| chemical compound, drug | Complete Protease Inhibitor Cocktail, EDTA-free | Sigma | Cat #5056489001 |
| chemical compound, drug | Phenylmethylsulfonylfluoride | Sigma | Cat #78830 |
| software, algorithm | R 3.3.0 | R foundation for Statistical Computing | www.R-project.org/ |
| software, algorithm | ggplot2 R package | Hadley Wickham | www.ggplot2.org/ |
| software, algorithm | reshape2 R package | Hadley Wickham | www.CRAN. R-project.org/ package=reshape2 |
| software, algorithm | gridExtra R package | Baptiste Auguie | www.CRAN. R-project.org/ package=gridExtra |
| software, algorithm | readr R package | Hadley Wickham, Jim Hester, Roman Francois | www.CRAN. R-project.org/ package=readr |
| software, algorithm | plyr R package | Hadley Wickham | www.CRAN. R-project.org/ package=plyr |
| software, algorithm | Cairo R package | Simon Urbanek and Jeffrey Horner | www.CRAN. R-project.org/ package=Cairo |
| software, algorithm | minpack.lm R package | Timur V. Elzhov, Katharine M. Mullen, Andrej-Nikolai Spiess, Ben Bolker | www.CRAN. R-project.org/ package=minpack.lm |
| software, algorithm | Python 3.5.2 | Python Software Foundation | www.python.org/ |
| software, algorithm | scipy Python package | Open source | www.scipy.org/ |
| software, algorithm | numpy Python package | Open source | www.numpy.org/ |
| software, algorithm | scikit-image Python package | Open source | www.scikit-image.org/ |
| software, algorithm | pyto_segmenter Python package | This paper | https://github. com/deniclab/ pyto_segmenter |
| software, algorithm | Various Python and R analysis scripts | This paper | https://github. com/deniclab/Weir_ 2017_Analysis |

## Yeast strain construction

All *S. cerevisiae* gene deletion and tagged strains were constructed using standard homologous recombination methods (*Longtine et al., 1998*) and are listed in the Key resources table. Cassettes

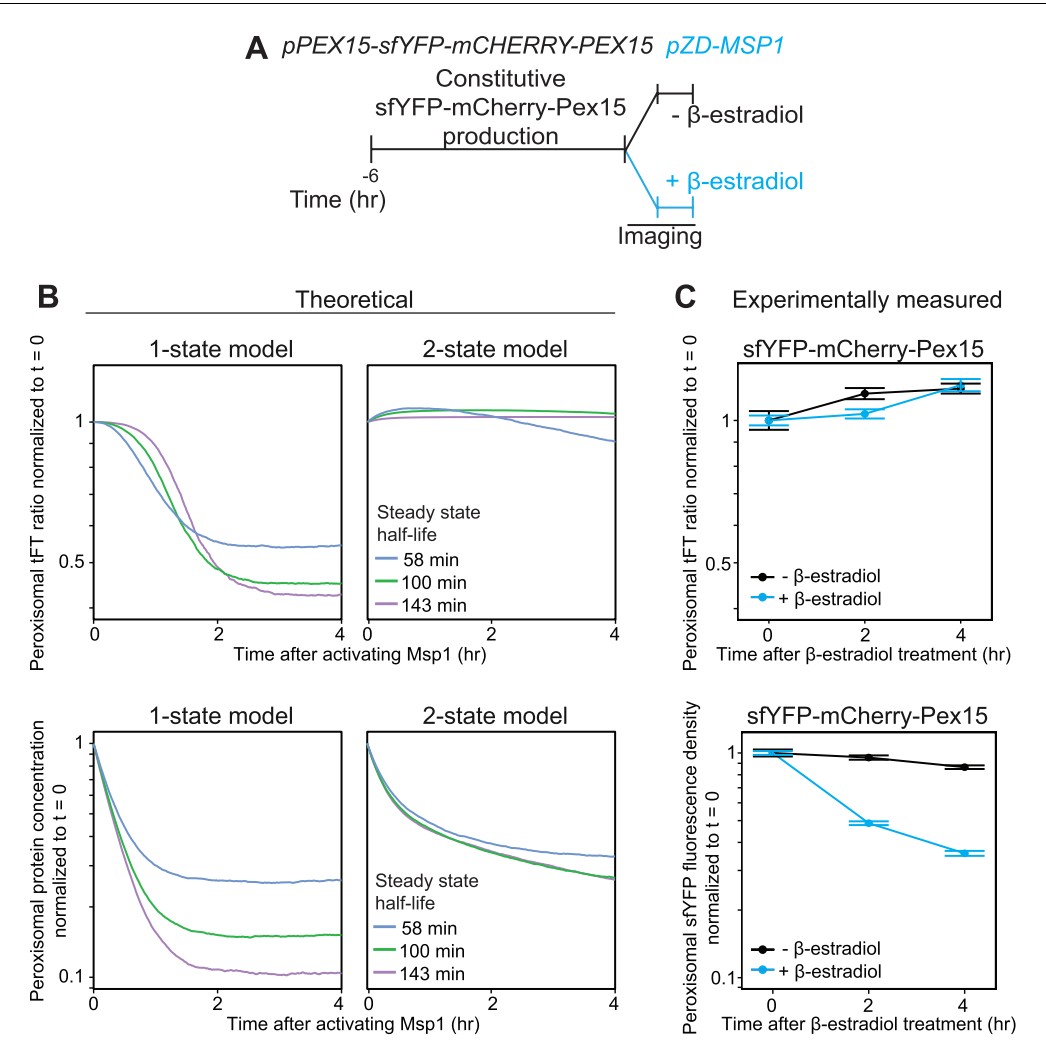

**Figure 5.** Experimental measurement of Pex15 levels and age after Msp1 activation compared to theoretical modeling. (**A**) Schematic of the staged expression experiment for monitoring turnover of sfYFP-mCherry-Pex15 expressed from the native genomic *PEX15* promoter following Msp1 overexpression. (**B**) Simulations of mCherry/ sfYFP ratio (top) and Pex15 decay (bottom) as a function of time following Msp1 activation in a 1-state or 2-state regime. Different colors show Pex15 dynamics resulting from the indicated theoretical half-life parameters. See Materials and methods for simulation details. (**C**) Experimentally measured mCherry/sfYFP ratio (top) and sfYFP density decay (bottom) from the experiment described in part *B*. Error bars represent standard error of the mean. Greater than 100 cells were imaged for each sample at each time point, and different fields of cells were imaged at each time point to minimize photobleaching. This experiment was performed twice with similar results.
DOI: https://doi.org/10.7554/eLife.28507.014

The following figure supplement is available for figure 5:

**Figure supplement 1.** Supporting information for tandem fluorescent timer-tagged Pex15.
DOI: https://doi.org/10.7554/eLife.28507.015

for fluorescent protein tagging at genes' endogenous loci were PCR amplified from the pKT vector series (*Sheff and Thorn, 2004*). Tandem fluorescent timer-tagged Pex15 was expressed from a transgene integrated at the *ura3* locus. Fluorescent peroxisome markers, expressed as transgenes from the *TRP1* locus, were generated by creating pKT plasmid variants containing the *S. cerevisiae TDH3* promoter upstream of a gene encoding a fluorescent protein with an engineered PTS1 sequence (Serine-Lysine-Leucine-stop). Strains with β-estradiol-induced Msp1 expression were made by homologous recombination of a 5' *LEU2*-marked Z4EV expression cassette with a 3' Z4EV-driven

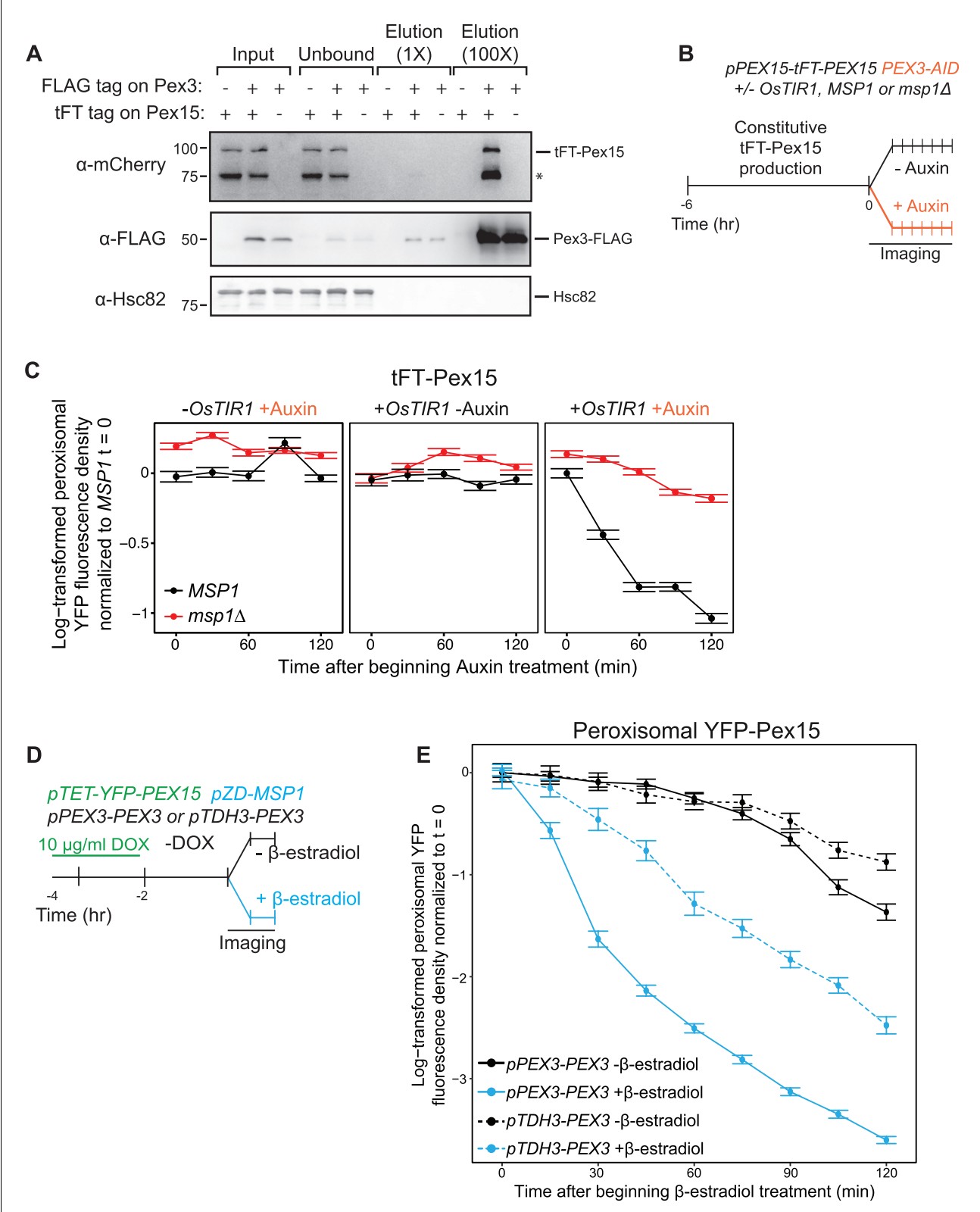

**Figure 6.** Pex3 protects Pex15 from Msp1-induced turnover. (**A**) Whole cell lysates prepared from *PEX3 msp1Δ* and *PEX3-FLAG msp1Δ* cells expressing tFT-Pex15 from its native genomic promoter or lacking tagged Pex15 were solubilized with NP-40 and then incubated with anti-FLAG resin. The resulting immunoprecipitates (IP) were washed and then eluted with FLAG peptide. IP inputs, post-IP depleted lysates, and IP elutions were resolved by SDS-PAGE and analyzed by immunoblotting with the indicated antibodies (see Materials and methods for details). *, a tFT-Pex15 degradation product

*Figure 6 continued on next page*

*Figure 6 continued*

lacking the N-terminal sfYFP. (B) Schematic of the staged degradation experiment for monitoring tFT-Pex15 turnover following Pex3-AID degradation. Wild-type and *msp1Δ* cells containing tFT-Pex15 constitutively expressed from the *PEX15* promoter and expressing Pex3-AID were grown in exponential phase for 6 hr. The experiment was performed in the presence and absence of the E3 ligase OsTir1 which ubiquitinates Pex3-AID following Auxin treatment (*Nishimura et al., 2009*). Half of the cells were then subjected to treatment with 1 mM Auxin while the other half received DMSO vehicle, followed by time-lapse imaging of both cell populations using a spinning disk confocal microscope. (C) Quantitation of peroxisomal sfYFP fluorescence from tFT-Pex15 from the experiment described in part *B*. YFP signal density at peroxisomes was determined as described in *Figure 3C* and plotted after normalization to the 0 hr timepoint of the identically treated *MSP1* strain. Error bars represent standard error of the mean. These data represent analysis of >100 cells for each sample at each timepoint. Different fields of cells were imaged at each timepoint to minimize photobleaching. This experiment was performed twice with similar results. (D) Schematic of the staged expression experiment for monitoring Msp1-dependent turnover of peroxisomal Pex15 in the presence and absence of overexpressed Pex3. This experiment was performed twice with similar results. (E) Quantitation of peroxisomal YFP-Pex15 fluorescence from the experiment described in part *D*. YFP signal density at peroxisomes was determined as described in *Figure 3C* and plotted as in *Figure 4F*. Pex3-overexpressing cells (*pTDH3-PEX3*) are shown with dashed lines, whereas solid lines indicate peroxisomal YFP levels in cells producing Pex3 from its endogenous promoter. These data represent analysis of 243 mock-treated and 128 β-estradiol-treated *PEX3* wild type cells and 171 mock-treated and 197 β-estradiol-treated *pTDH3-PEX3* cells followed throughout the time course as well as progeny from cell divisions during the experiment.

DOI: https://doi.org/10.7554/eLife.28507.016

The following figure supplement is available for figure 6:

**Figure supplement 1.** Supporting evidence for Pex15 interaction with Pex3 and rapid in situ destruction of Pex3-AID.

DOI: https://doi.org/10.7554/eLife.28507.017

(*ZD*) promoter (*McIsaac et al., 2013*) upstream of the endogenous *MSP1* ORF. Similar cassettes were constructed for yeast expression of Pex22$_{1-35}$-Msp1$_{32-362}$ protein and Msp1$_{1-12}$-Tom70$_{12-29}$-Msp1$_{28-362}$ from the endogenous *MSP1* locus. Strains with doxycycline-induced expression of Pex15 variants were made by homologous recombination of a 5' *CgTRP1*-marked expression cassette the G76V variant of the reverse tetracycline transactivator (*rTA*) (*Roney et al., 2016*) with a 3' *GAL1* promoter variant altered for control by rTA driving expression of the *YFP* ORF (lacking a stop codon) fused to the *PEX15* ORF or mutant variant, and followed by the *PEX15* terminator. This cassette was integrated into the *ura3* locus of strains as indicated in the strain table. *PEX3-FLAG* was generated by integrating a previously described C-terminal 3 × FLAG tagging cassette (*Denic and Weissman, 2007*).

## Immunoblotting analysis

Yeast cultures were grown overnight to 0.8 OD$_{600}$ units at 30°C in YEPD (1% yeast extract (BD Biosciences, San Jose, CA), 2% bacto-peptone (BD Biosciences), 2% glucose (Sigma, St. Louis, MO)) and treated with 3-indoleacetic acid (auxin, 500 µM) (Sigma), cycloheximide (100 µg/mL) (Sigma) or DMSO vehicle as indicated. Cells were pelleted by 3000 × g centrifugation for 1 min, resuspended in ice cold 0.2 M NaOH and incubated on ice for 10 min. Cells were then pelleted by 10,000 × g centrifugation for 1 min and boiled in SDS-PAGE sample buffer (50 mM Tris-HCl pH 6.8, 2.5% sodium dodecyl sulfate, 0.008% bromophenol blue, 10% glycerol, 5% β-mercaptoethanol). Following centrifugation to remove any insoluble cell debris, supernatant samples were resolved by SDS-PAGE (70 min at 195V) using Novex 4–20% Tris-Glycine gels (Thermo Fisher Scientific, Waltham, MA) and electroblotted onto nitrocellulose membranes. Blocking and antibody incubations (mouse anti-FLAG M2 (Sigma), mouse anti-V5 R960-25 (Thermo Fisher Scientific), mouse anti-GFP (Sigma), mouse anti-Pgk1 22C5D8 (Thermo Fisher Scientific), rabbit anti-Hsc82 ab30920 (Abcam), and rabbit anti-Sdh4 (gift of N. Pfanner)) were performed in 5% milk in TBST (10 mM Tris-HCl pH 7.4, 150 mM NaCl, 0.25 mM EDTA, 0.05% Tween-20). HRP-conjugated secondary antibodies (BioRad, Hercules, CA) were detected following incubation with SuperSignal West Femto Substrate (Thermo Fisher Scientific) using a ChemImager (AlphaInnotech, San Jose, CA). Fluorescent secondary antibodies (Thermo Fisher Scientific) were detected using a Typhoon Trio imager (GE Healthcare, Chicago, IL).

## Protease protection of YFP-Pex15$_{\Delta C30}$-V5 at mitochondria

VDY3412 cells were pre-grown to late log phase (1 OD$_{600}$) in 100 mL YEPD and then diluted to 0.1 OD$_{600}$ in 1 L YEPD. Cells were grown with shaking at 30°C to 1 OD$_{600}$ and then treated with 50 µg/ml doxycycline (Sigma) for 4 hr at 30°C with shaking. Cells were harvested by centrifugation. Crude

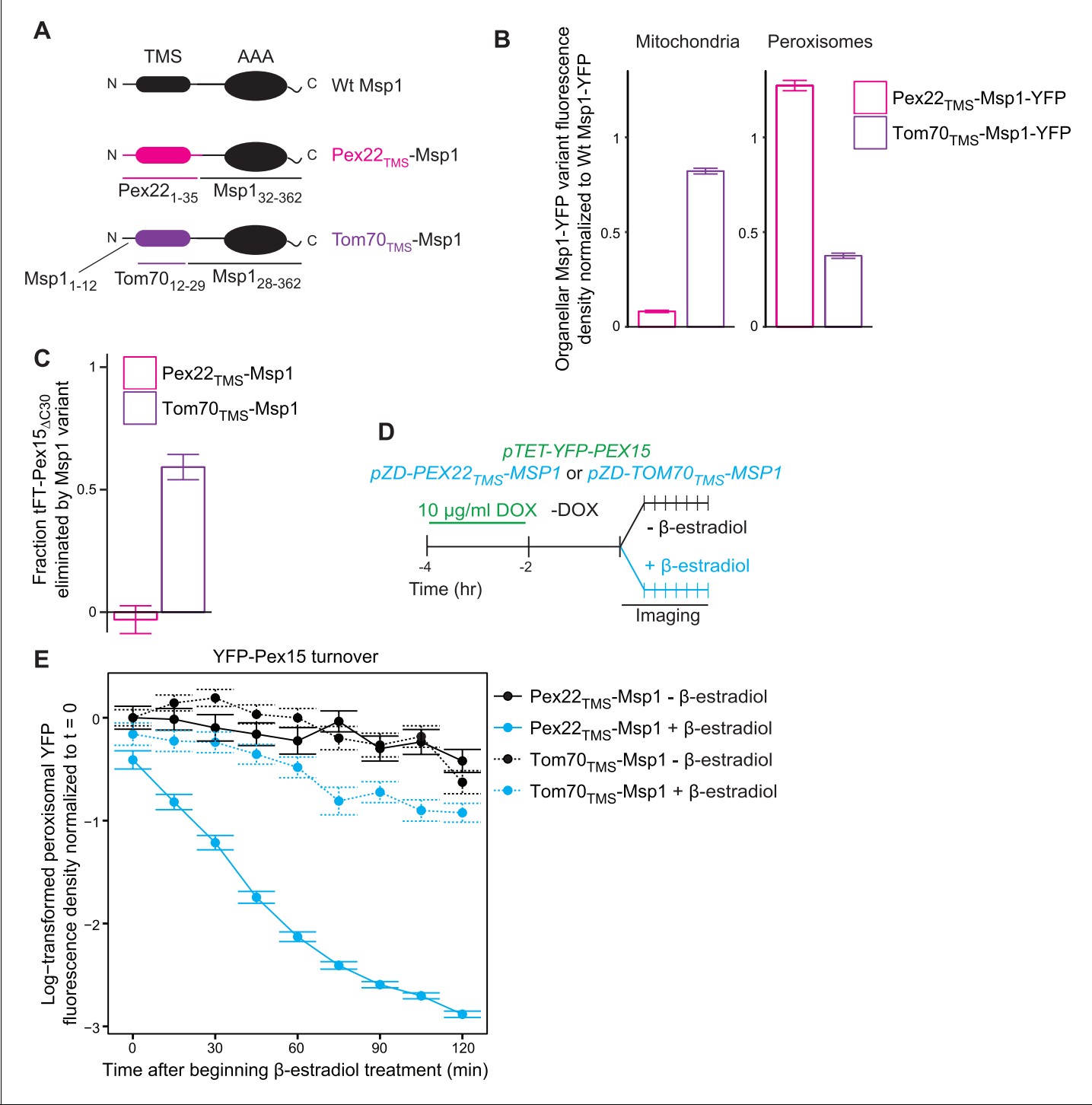

**Figure 7.** The Msp1 TMS targets Msp1 to peroxisomes and mitochondria, but is dispensable for substrate engagement. (**A**) Schematic representation of Msp1 TMS chimaeras. Top, wild type Msp1, with an N-terminal TMS and C-terminal AAA ATPase domain. Middle, Msp1 with its N-terminal 31 amino acids deleted and replaced with the first 35 amino acids of Pex22. Bottom, Msp1 with its TMS (residues 13–27) replaced with residues 12–29 of Tom70. (**B**) Quantitation of mitochondrial (left) and peroxisomal (right) YFP density in cells producing Pex22_{TMS}-Msp1-YFP (pink) or Tom70_{TMS}-Msp1-YFP (purple) from the native *MSP1* promoter. YFP signal density was determined at mitochondria as described in *Figure 2C*, and at peroxisomes as described in *Figure 3C*. YFP fluorescence density at each organelle was normalized to the mean fluorescence density at the same organelle for wild type Msp1-YFP. Error bars represent standard error of the population mean. See *Figure 7—figure supplement 1A* for representative images. These data represent analysis of >250 cells from each strain. (**C**) Quantitation of endogenously expressed mitochondrial tFT-Pex15_{ΔC30} sfYFP fluorescence density in cells producing Pex22_{TMS}-Msp1 or Tom70_{TMS}-Msp1 from the native *MSP1* promoter. Mitochondrial sfYFP fluorescence density was calculated

*Figure 7 continued on next page*

*Figure 7 continued*

as described for *Figure 3E*. Bars represent the fraction of tFT-Pex15$_{\Delta C30}$ eliminated by the Msp1 variant, calculated as (mitochondrial mean YFP density in *msp1Δ* - mitochondrial mean YFP density in TMS variant)/(mitochondrial mean YFP density in *msp1Δ* - mitochondrial mean YFP density in *MSP1*). See *Figure 7—figure supplement 1B* for a violin plot of the mitochondrial sfYFP fluorescence distributions. These data represent analysis of >250 cells from each strain. (D) Experimental timeline of the staged expression experiment for monitoring turnover of peroxisomal Pex15 following transient expression of Pex22$_{TMS}$-Msp1 or Tom70$_{TMS}$-Msp1 from the *ZD* promoter. (E) Quantitation of peroxisomal YFP-Pex15 fluorescence from the experiment described in part *D*. YFP signal density at peroxisomes was determined as described in *Figure 3C* and plotted as in *Figure 4F*. Cells producing Pex22$_{TMS}$-Msp1 and Tom70$_{TMS}$-Msp1 from the *ZD* promoter are shown with solid lines and dashed lines respectively. These data represent analysis of 158 mock-treated Pex22$_{TMS}$-Msp1 cells, 231 β-estradiol-treated Pex22$_{TMS}$-Msp1 cells, 130 mock-treated Tom70$_{TMS}$-Msp1 cells, and 171 β-estradiol-treated Tom70$_{TMS}$-Msp1 cells followed throughout the time course as well as progeny from cell divisions during the experiment.

DOI: https://doi.org/10.7554/eLife.28507.018

The following figure supplement is available for figure 7:

**Figure supplement 1.** Supporting evidence for the role of the Msp1 TMS in membrane targeting and substrate clearance.

DOI: https://doi.org/10.7554/eLife.28507.019

mitochondria were isolated from harvested cells as described previously (*Meisinger et al., 2006*). 100 µg of crude mitochondria was treated with 10 µg Proteinase K (Roche, Basel, Switzerland) or mock treated in the presence or absence of 1% Triton X-100 (Sigma) at room temperature for 30 min. Phenylmethanesulfonyl fluoride (PMSF) (Sigma) was added to each sample to a final concentration of 5 mM to inhibit Proteinase K and samples were incubated 10 min on ice. Samples were mixed with boiling SDS-PAGE sample buffer and subjected to SDS-PAGE and immunoblotting analysis as described earlier.

## Live-cell imaging of tagged Pex15 and Msp1

Cells were inoculated into 2 mL of complete synthetic media with glucose (0.67% yeast nitrogen base (BD Biosciences), 2% glucose, 1 × CSM (Sunrise Sciences, San Diego, CA)) and grown overnight at 30°C on a roller drum. The following morning, cells were back-diluted to 0.05 OD$_{600}$ in fresh media and grown to mid-to-late log phase (0.5–1 OD$_{600}$) for imaging with drug treatments as indicated in figure schematics. β-estradiol (Sigma) was used at 1 µM for all experiments; doxycycline was used at concentrations indicated in figure legends. Cells in culture media were applied directly to the well of a concanavalin A (MP Biomedicals, Santa Ana, CA)-coated Lab-Tek II chambered coverglass (Thermo Fisher) and allowed to adhere for 5 min at room temperature. Culture media was removed and adhered cells were immediately overlaid with a 1% agarose pad containing complete synthetic media with glucose and supplemented with drugs when applicable. The agarose pad was

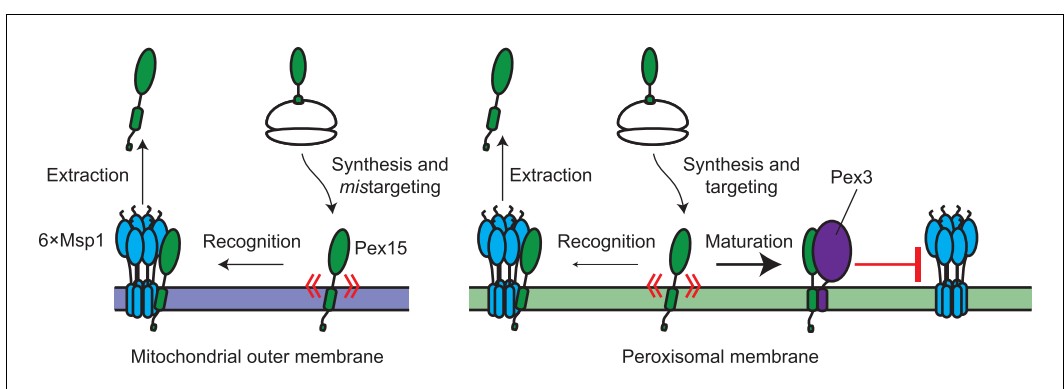

**Figure 8.** Model for substrate selection by Msp1. On the left, mistargeted Pex15 inserts into mitochondria and is then recognized by Msp1 for extraction. On the right, following insertion into peroxisomes, nascent Pex15 can be recognized by Msp1 in principle but in practice this requires either Pex15 and/or Msp1 to be present above their usual levels. Otherwise, normal Msp1 recognition is slow relative to the faster 'maturation' process involving Pex3 interaction with Pex15, which blocks Msp1 recognition.

DOI: https://doi.org/10.7554/eLife.28507.020

overlaid with liquid media for timelapse imaging experiments. Live-cell imaging was performed at 25°C on a TI microscope (Nikon, Tokyo, Japan) equipped with a CSU-10 spinning disk (Yokogawa, Tokyo, Japan), an ImagEM EM-CCD camera (Hamamatsu, Hamamatsu, Japan), and a 100 × 1.45 NA objective (Nikon). The microscope was equipped with 447 nm, 515 nm and 591 nm wavelength lasers (Coherent, Santa Clara, CA) and was controlled with MetaMorph imaging software (Molecular Devices, Sunnyvale, CA). Z-stacks were acquired with 0.2 µm step size for 6 µm per stack. Camera background noise was measured with each Z-stack for normalization during timelapse imaging.

### Sample size estimation and experimental replication details

For quantitative microscopy experiments, the number of cells present in each sample was manually counted in brightfield images and indicated in the associated figure legend. Each experiment was repeated the number of times indicated in the associated figure legend. Replicates represent technical replicates in which the same strains were subjected to repetition of the entire experiment, often on different days.

### Image post-processing and organelle segmentation

All fluorescence images were normalized to background noise to compensate for uneven illumination and variability in camera background signal. To identify peroxisomes and mitochondria, images of their respective markers were processed by an object segmentation script. Briefly, images were smoothed using a Gaussian filter and then organelle edges were identified by processing each slice with a Canny edge detector (**Canny, 1986**) implemented in the Python package scikit-image. Enclosed objects were filled and individual three-dimensional objects were identified by locally maximizing Euclidean distance to the object border. Individual objects were identified and separated by watershed segmentation as implemented in scikit-image. For mitochondria, contiguous but separately segmented objects were merged to form one mitochondrion. For YFP-Pex15 quantitation at mitochondria, regions of mitochondria that overlapped with peroxisomes were removed by eliminating segmented mitochondria pixels that overlapped with segmented peroxisomes. Segmentation code is available at http://www.github.com/deniclab/pyto_segmenter (**Weir, 2017a**) and sample implementation is available at www.github.com/deniclab/Weir_2017_analysis (**Weir, 2017b**) (copies archived at https://github.com/elifesciences-publications/pyto_segmenter and https://github.com/elifesciences-publications/Weir_2017_analysis respectively). Raw source images are available on the Dryad data repository associated with this manuscript.

### Fluorescence intensity analysis

Following organelle segmentation, total fluorescence intensity for Pex15 was determined in each segmented object by summing intensities in the corresponding pixels for YFP fluorescence images (and mCherry images for mCherry-sfYFP-Pex15 and mCherry-sfYFP-Pex15$_{\Delta C30}$ in **Figure 5C**). Fluorescence density was calculated by dividing total pixel intensity by object volume in pixels. Background was calculated empirically by measuring Pex15 fluorescence intensity in peroxisomes and/or mitochondria in cells lacking fluorescently labeled Pex15, and the mean background density was subtracted from each segmented object's fluorescence density. Because Pex15 fluorescence density was approximately log-normally distributed, mean and standard error of the mean were calculated on logarithmically transformed fluorescence densities when applicable. Plotting was performed using R and the ggplot2 package. See www.github.com/deniclab/Weir_2017_analysis for tabulated data and analysis code.

### Model fitting and statistics

For 1-state and 2-state model fitting, organelle fluorescence density means were first normalized to the sample's mean at time 0. For the 1-state model, log-transformed mean fluorescence densities at each time point were fit to a linear model using least squares fitting in R. For the 2-state model, logarithmically transformed data was fit to a logarithmically transformed version of a previously derived 2-state degradation model (**Sin et al., 2016**) using the Levenberg-Marquardt algorithm (**Levenberg, 1944**) for non-linear least squares fitting as implemented in the R package minpack.lm. Error for fit parameters was obtained from fit summary statistics. The difference between the 1-state and

2-state model fits was determined by integrating the difference between the two fit equations over the measured time interval, then dividing by the time interval to normalize across timecourse experiments of different lengths. See www.github.com/deniclab/Weir_2017_analysis for tabulated data and R code. Observed half-life was determined by converting the peroxisomal YFP-Pex15 –Msp1 $k_{decay}$ (*Figure 4—figure supplement 1*) using the equation half-life = ln(2)/$k_{decay}$, and then multiplied by 60 to convert from hours to minutes. Error bars represent standard error of the mean.

## Simulation of protein age and turnover

To stochastically model peroxisomal Pex15 levels and age following transient Msp1 expression, we used a Gillespie algorithm approach (*Gillespie, 1977*). In brief, this approach cycles through the following steps: 1. Model the expected time until the next 'event' takes place (import, degradation, or maturation of a Pex15 molecule) by summing event rates and drawing from an exponential distribution based on the summed rate constant, 2. Age all simulated Pex15 molecules according to time passage, 3. Determine which of the possible events took place by weighted random draws based on each event's probability of occurring, 4. Execute that event, and then repeat these steps until the simulation's time has expired. Based on our observation that Pex15 turnover in the absence of Msp1 occurs with exponential decay kinetics (*Figure 4F*), we established starting conditions by drawing 1000 ages from an exponential distribution with half-life indicated in *Figure 5B*. For the rest of the simulation we used this rate constant to predict import of new molecules and as a steady-state degradation rate constant (and as $k_{decay,2}$ in 2-state simulations). We treated this vector of 1000 ages as a single peroxisome containing 1000 Pex15 molecules (this is likely an over-estimation of Pex15 amounts in many cases, but over-estimating Pex15 levels improved statistical robustness of the analysis and did not alter simulation mean outcomes). When simulating steady state 2-state behavior using the calculated $k_{mat}$ value, we found that ~60% of the elements existed in the 'unstable' form at steady state (data not shown) and therefore used this as a starting value. For 2-state simulations we randomly drew 600 of the vector elements to be 'unstable' at the start of the simulation, weighting probabilities of each draw using an exponential distribution with $k_{mat}$ as the decay rate constant. After validating that our starting conditions represented a stable steady state by simulating without perturbing rate constants, we began the reported simulations with $k_{decay}$ set to 2.82 $hr^{-1}$, the best linear fit for turnover from the first three time points (for 1-state simulations), or with $k_{decay,1}$ (for 2-state simulations) set to the calculated value from *Figure 4F*. Simulations ran for 4 hr of simulated time and values for particle age and abundance were recorded at every simulated minute. 100 simulations were performed with each set of parameters and the mean particle age and abundance at each minute were calculated across the 100 simulations. Finally, we modeled maturation of sfYFP fluorescence and mCherry fluorescence based on established maturation half-times (*Hansen and O'Shea, 2013*; *Khmelinskii et al., 2012*), respectively) and calculated the mean population tFT ratio at each minute. We normalized these data to the value at the simulation's starting point. See the www.github.com/deniclab/Weir_2017_analysis for Gillespie simulation R code.

## Pex3-GFP-AID fluorescence microscopy

Yeast cultures were grown overnight in synthetic medium to 0.5 $OD_{600}$ and treated with 3-indoleacetic acid (Auxin, 1 mM) (Sigma) or DMSO vehicle as indicated. Following concentration of cells by centrifugation, cells were imaged at room temperature on an Axiovert 200M microscope body (Carl Zeiss, Oberkochen, Germany) equipped with a CSU-10 spinning disk (Yokogawa) and 488 nm and 561 nm lasers (Coherent) using an oil-immersion 100 × 1.45 NA objective (Carl Zeiss). Images were acquired using a Cascade 512B EM-CCD detector (Photometrics, Tuscon, AZ) and MetaMorph acquisition software (Molecular Devices).

## Pex3-FLAG immunoprecipitation

1 L yeast cell culture was grown to 1.8–2.2 $OD_{600}$ in YEP +5% glucose at 30°C with shaking. Cells were collected by centrifuging 20 min at 3000 × g, 4°C, then washed once with 50 ml sterile $H_2O$. Cells were resuspended in 1 ml ice-cold lysis buffer (50 mM HEPES-KOH pH 6.8, 150 mM KOAc, 2 mM $MgCl_2$, 1 mM $CaCl_2$, 0.2 M sorbitol, 2x cOmplete protease inhibitors (Sigma)) per 6 g wet weight, and dripped into liquid nitrogen to flash-freeze. Cells were lysed cryogenically using a PM100 ball mill (Retsch, Haan, Germany) and stored at −80°C. 0.4 g lysed cell powder was thawed

on ice and mixed with 1.6 mL IP buffer (50 mM HEPES-KOH pH 6.8, 150 mM KOAc, 2 mM Mg [OAc]$_2$, 1 mM CaCl$_2$, 15% glycerol, 1% NP-40, 5 mM sodium fluoride, 62.5 mM β-glycerophosphate, 10 mM sodium vanadate, 50 mM sodium pyrophosphate). Lysates were detergent solubilized at 4°C for 1 hr with nutation and then subjected to low-speed centrifugation (twice at 3000 × g, 4°C for 5 min) to remove any unlysed cells and cell debris. The supernatants were further cleared by ultracentrifugation (100,000 × g, 4°C for 30 min) before adding 40 µL protein G Dynabeads (Sigma) conjugated to anti-FLAG M2 monoclonal antibody (Sigma). Following incubation for 3 hr at 4°C with nutation, Dynabeads were washed four times with IP buffer and bound proteins were eluted at room temperature with two sequential rounds of 10 µl 1 mg/mL 3 × FLAG peptide (Sigma) in IP buffer. Immunoblotting analysis was performed as described above.

## Note added in proof

A complementary structure-function analysis of Msp1 was published while this work was under review (*Wohlever et al., 2017*).

## Acknowledgements

We thank A Murray, E O'Shea, D Botstein, S McIsaac, N Pfanner, and A Amon for reagents, S Mukherji for modeling advice, L Bagamery for microscopy assistance, and members of the Denic Laboratory, M Gropp, A Murray, and R Gaudet for comments on the manuscript. This work was supported by the National Institutes of Health (R01GM099943-04).

## Additional information

### Funding

| Funder | Author |
|---|---|
| National Institutes of Health | Vladimir Denic |

The funders had no role in study design, data collection and interpretation, or the decision to submit the work for publication.

### Author contributions

Nicholas R Weir, Data curation, Software, Formal analysis, Investigation, Visualization, Methodology, Writing—original draft, Writing—review and editing; Roarke A Kamber, James S Martenson, Investigation, Visualization, Writing—review and editing; Vladimir Denic, Conceptualization, Supervision, Funding acquisition, Methodology, Writing—original draft, Project administration, Writing—review and editing

### Author ORCIDs

Nicholas R Weir, http://orcid.org/0000-0002-1797-849X
Vladimir Denic, http://orcid.org/0000-0002-1982-7281

### Decision letter and Author response

Decision letter https://doi.org/10.7554/eLife.28507.024
Author response https://doi.org/10.7554/eLife.28507.025

## Additional files

### Supplementary files

• Transparent reporting form
DOI: https://doi.org/10.7554/eLife.28507.021

### Major datasets

The following dataset was generated:

| Author(s) | Year | Dataset title | Dataset URL | Database, license, and accessibility information |
|---|---|---|---|---|
| Weir NR, Kamber RA, Martenson JS, Denic V | 2017 | Data from: The AAA protein Msp1 mediates clearance of excess tail-anchored proteins from the peroxisomal membrane | http://dx.doi.org/10.5061/dryad.pc4d6 | Available at Dryad Digital Repository under a CC0 Public Domain Dedication |

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
