## [Decision Letter]

Thank you for submitting your article "A New Quantitative Cell Microscopy Approach Reveals Mechanistic Insights into Clearance of Membrane Substrates by Msp1" for consideration by *eLife*. Your article has been reviewed by three peer reviewers, one of whom is a member of our Board of Reviewing Editors and the evaluation has been overseen by Ivan Dikic as the Senior Editor. The reviewers have opted to remain anonymous.

The reviewers have discussed the reviews with one another and the Reviewing Editor has drafted this decision to help you prepare a revised submission.

Summary:

This study has investigated the function of the AAA+ ATPase Msp1, a recently discovered factor needed for degradation of tail–anchored (TA) membrane proteins mislocalized to mitochondria and peroxisomes. Relatively little is known about what defines an Msp1 substrate or how this factor facilitates substrate degradation, but this pathway is likely to be important for maintenance of mitochondrial (and presumably peroxisomal) homeostasis. Earlier work had established that the peroxisomal protein Pex15 is a target for Msp1–mediated degradation when it is mis–targeted to mitochondria. Yet, Pex15 can co–exist with Msp1 in peroxisomes. Why Pex15 is a target for the Msp1 pathway in mitochondria but not in peroxisomes is the main issue that has been addressed. The central proposal in this study is that Msp1–dependent degradation of Pex15 occurs only when Pex15 is free from its Pex3 binding partner. The primary support for this conclusion is the observation that Pex15 over–expression, which presumably saturates its Pex3 partner, results in Msp1–dependent Pex15 degradation even from peroxisomes. The implication is that Msp1 may select its clients on the basis of being 'solitary' from their native multi–protein complexes. This would explain how Msp1 can reside in both mitochondria and peroxisomes to degrade only mislocalized proteins while avoiding normal residents.

Essential revisions:

The main new suggestion in this manuscript, that Pex15 in isolation from its Pex3 binding partner is the molecular target of Msp1 in peroxisomes, was found to be plausible but not strongly established. Thus, most of the suggested experiments are aimed toward improving support for this key claim.

1) While the model presented is intriguing, the data do not, at present, fully support the claim that Msp1 is active on the peroxisomal membrane. The authors claim that Pex3 protects Pex15 from Msp1 mediated degradation. However, there is no experimental indication that this interaction occurs on the peroxisomal membrane. In fact, in Lam et al., 2010, Pex3 was shown to interact with Pex15 only during biogenesis and in the ER. Thus, to establish the central claim of this study, it seems critical to convincingly show that Pex15 actually interacts with Pex3 in peroxisomes, thereby lending support to the idea that this interaction is the protective agent in peroxisomes.

2) Part of the support for Pex3 being important in protecting Pex15 from degradation is the finding that Pex3 deletion results in Pex15 degradation. However, controls are not shown to ascertain whether the effect of Pex3 elimination causes a change in the peroxisomal membrane that alters the half–life of all peroxisomal proteins. It is important to establish the degree of specificity of Msp1–mediated degradation upon Pex3 deletion and rule out a more trivial generic destabilization of peroxisomes in general.

3) The authors claim that the degraded Pex15 in peroxisomes is the overexpressed fraction that presumably saturates Pex3. This claim would be more strongly supported by three additional results. First (and easiest), it is important to establish that the low concentration of Doxycycline used in Figure 3 (for example) causes overexpression of the protein. The authors could, for example, add a control image that compares the overexpressed phenotype with its native expression levels to be sure that the experiment starts when Pex15 is actually overexpressed. Second, it is worth testing whether co–expression of Pex3 can blunt Msp1 degradation of Pex15. Third, it is worth testing whether quantitative IP of Pex3 is able to co–deplete Pex15 in wt cells, that excess Msp1 has no effect on this, and that co–depletion is only partial when Pex15 is overexpressed (i.e., showing that it is indeed in excess of Pex3).

4) The experiment showing that Pex15 is actually inserted into the membrane as a transmembrane protein was not convincing (and I'm a bit suspicious about whether this is actually the case). The C–terminal region does not seem to contain a transmembrane segment by prediction algorithms (e.g., TMHMM), and manual inspection fails to reveal any region that contains a continuous hydrophobic sequence longer than ~10 residues. It is in this context that the protease protection experiment shown in Figure 2 becomes important. Unfortunately, the blot is rather dirty in the critical control lane. This appears to be due to Triton X–100 in that sample forming mixed micelles with SDS that migrate near and distort the region of the blot where their key protected fragment runs. This distortion makes it quite plausible that the band seen with detergent might well be the same band seen without detergent, but altered in its migration due to the mixed micelle artifact. If this were the case, the data would not convincingly support transmembrane insertion. Thus, the authors must clean up this experiment. There are a couple of options: (i) use far less detergent, such as 0.1% Tx100, to minimize the artifact; (ii) immunoprecipitate the sample before doing the blot, with the last wash of the IPs being in detergent–free buffer to avoid this artifact.

---

## [Author Response]

Essential revisions:The main new suggestion in this manuscript, that Pex15 in isolation from its Pex3 binding partner is the molecular target of Msp1 in peroxisomes, was found to be plausible but not strongly established. Thus, most of the suggested experiments are aimed toward improving support for this key claim.1) While the model presented is intriguing, the data do not, at present, fully support the claim that Msp1 is active on the peroxisomal membrane. The authors claim that Pex3 protects Pex15 from Msp1 mediated degradation. However, there is no experimental indication that this interaction occurs on the peroxisomal membrane. In fact, in Lam et al., 2010, Pex3 was shown to interact with Pex15 only during biogenesis and in the ER. Thus, to establish the central claim of this study, it seems critical to convincingly show that Pex15 actually interacts with Pex3 in peroxisomes, thereby lending support to the idea that this interaction is the protective agent in peroxisomes.

We appreciate the work done by Lam et al., 2010 but our reading of their data led us to a different conclusion. In that manuscript, the authors isolated microsomes from yeast cells and then performed vesicle budding assays to produce vesicles containing both Pex3 and Pex15 (distinct from COPII vesicles that can co–bud from microsomes under these in vitro conditions). However, these experiments do not demonstrate any evidence of physical interaction between Pex3 and Pex15 (in the ER or in the aforementioned vesicles) nor is this one of the claims by the authors. Indeed, the only experiment performed to assess whether Pex3 and Pex15 directly interact is shown in Figure 5 of that paper, in which the authors affinity–purified Pex15 from budded vesicles in the presence or absence of Triton X–100. A small fraction of Pex3 only copurified with Pex15 in the absence of detergent, suggesting that these proteins reside in the same vesicles to some extent, but do not physically interact (or at least not in a way that persists following immunoprecipitation in the presence of detergent). Further contributing to this confusion, the Saccharomyces Genome Database has misannotated the existence of a Pex3 and Pex15 reconstituted complex based on this manuscript, when in fact the data only support cofractionation in vesicles.

Related to this matter, we thank reviewer #2 for pointing out that a bulk–cell interaction between Pex3 and Pex15 was previously observed using split ubiquitin technology in Eckert and Johnsson, 2003 (now referenced in the revised manuscript). This inspired us to use a related approach with sub–cellular resolution to experimentally test our claim that Pex15 and Pex3 physically interact at peroxisomes. Specifically, we made yeast strains expressing from endogenous loci Pex15 and Pex3 tagged with the N– and C–terminal fragments of the fluorescent protein Venus, respectively. Only in the presence of both fluorescent protein fragment tags did we observe bimolecular fluorescence complementation resulting in Venus fluorescence, which co–localized fluorescence from a peroxisomal matrix marker (see Author response image 1).

2) Part of the support for Pex3 being important in protecting Pex15 from degradation is the finding that Pex3 deletion results in Pex15 degradation. However, controls are not shown to ascertain whether the effect of Pex3 elimination causes a change in the peroxisomal membrane that alters the half–life of all peroxisomal proteins. It is important to establish the degree of specificity of Msp1–mediated degradation upon Pex3 deletion and rule out a more trivial generic destabilization of peroxisomes in general.

This concern is well taken and we have now tested the degree of specificity of Msp1–sensitivity among several other peroxisomal membrane proteins upon Pex3 depletion. Before addressing this concern, we reasoned that our original data might underestimate the actual contribution of Pex3 to protection of natively–expressed Pex15 because they were obtained following TET inducible overexpression of YFP–Pex15. Thus, we switched to monitoring the dynamics of peroxisomal membrane proteins expressed from their native promoters. Indeed, this resulted in a more dramatic destabilization of Pex15 by an Msp1–dependent process (see Figure 6 compared to Figure 6—figure supplement 1 in the revised manuscript). By contrast, two other peroxisomal membrane proteins, Pex11 and Pex12, persisted at peroxisomes following degradation of Pex3, regardless of whether Msp1 was present or not (Figure 6—figure supplement 1 in the revised manuscript). Taken together, these new data argue against the possibility that Pex3 depletion induces generic Msp1–dependent destabilization of peroxisomal membrane proteins.

3) The authors claim that the degraded Pex15 in peroxisomes is the overexpressed fraction that presumably saturates Pex3. This claim would be more strongly supported by three additional results. First (and easiest), it is important to establish that the low concentration of Doxycycline used in Figure 3 (for example) causes overexpression of the protein. The authors could, for example, add a control image that compares the overexpressed phenotype with its native expression levels to be sure that the experiment starts when Pex15 is actually overexpressed. Second, it is worth testing whether co–expression of Pex3 can blunt Msp1 degradation of Pex15. Third, it is worth testing whether quantitative IP of Pex3 is able to co–deplete Pex15 in wt cells, that excess Msp1 has no effect on this, and that co–depletion is only partial when Pex15 is overexpressed (i.e., showing that it is indeed in excess of Pex3).

We thank the reviewers for these suggestions, which also expand on reviewer #3’s minor points. In the original Figure 3—figure supplement 1 we compared the peroxisomal levels of YFPPex15 produced from its endogenous promoter to the amount produced following 2 hours of induction with 5 ug/ml doxycycline, a concentration lower than that used for any of our Pex15 turnover experiments. This resulted in YFP–Pex15 levels 10 times higher at peroxisomes than in cells where the protein was produced from the endogenous *PEX15* promoter. To test if elevated expression of Pex3 could blunt Msp1 degradation of excess Pex15 molecules, we overexpressed Pex3 from the strong *TDH3* promoter and obtained the expected result. Since this approach didn’t confer complete protection from Msp1, it points to the presence of additional protection factors. Indeed, Pex15 has been shown to interact with several other peroxisomal membrane proteins in previous studies (for examples, see Eckert and Johannson, 2003 and Rosenkranz et al.,2006, among others), which may play a role in shielding Pex15 from Msp1. We have added a section to the discussion covering these points.

Lastly, we have attempted to quantitate the extent of the Pex3–Pex15 interaction after extraction from membranes with detergent. After optimizing our affinity purification of Pex3 (natively expressed), we found that near–quantitative Pex3 from the flow–through was accompanied by only a partial depletion of Pex15 (also natively expressed). Since affinity purification protocols in detergent have a tendency to disrupt labile native membrane protein interactions, we plan to develop in the future quantitative in vivo reporters of Pex3–Pex15 complex dynamics (e.g. FRET–based).

4) The experiment showing that Pex15 is actually inserted into the membrane as a transmembrane protein was not convincing (and I'm a bit suspicious about whether this is actually the case). The C–terminal region does not seem to contain a transmembrane segment by prediction algorithms (e.g., TMHMM), and manual inspection fails to reveal any region that contains a continuous hydrophobic sequence longer than ~10 residues. It is in this context that the protease protection experiment shown in Figure 2 becomes important. Unfortunately, the blot is rather dirty in the critical control lane. This appears to be due to Triton X–100 in that sample forming mixed micelles with SDS that migrate near and distort the region of the blot where their key protected fragment runs. This distortion makes it quite plausible that the band seen with detergent might well be the same band seen without detergent, but altered in its migration due to the mixed micelle artifact. If this were the case, the data would not convincingly support transmembrane insertion. Thus, the authors must clean up this experiment. There are a couple of options: (i) use far less detergent, such as 0.1% Tx100, to minimize the artifact; (ii) immunoprecipitate the sample before doing the blot, with the last wash of the IPs being in detergent–free buffer to avoid this artifact.

The relatively hydrophilic transmembrane segments of mitochondrial and peroxisomal membrane proteins are notoriously difficult to predict by algorithms trained using databases dominated by the more hydrophobic secretory pathway membrane proteins. For example, TMHMM does not predict that either Msp1 or Pex14 are membrane–integral even though this is well–established by protease protection assays (see Chen et al.,2014 and Albertini et al.,1997, respectively). Thus, we took the advice to improve the quality of our own protease protection assay for Pex15ΔC30 in earnest. We repeated this assay using a room temperature protease treatment rather than 4 °C and did not observe any background banding in the Triton X–100 control lane (see Figure 2). We note that room temperature incubation of mitochondria resulted in loss of the N–terminal YFP from most molecules of Pex15ΔC30 as well as Proteinase K–independent degradation of Tom70. We therefore did not include the anti–Tom70 control blot. A related concern was expressed by reviewer #2 about protein size, prompting us to now display the contiguous anti–V5 blot (with molecular markers indicated) that shows full–length and YFP clipped Pex15ΔC30–V5, as well as the much smaller protected fragment (PF) that has the expected size of the Pex15 transmembrane domain–V5.